# Visualizing the multi-level assembly structures of conjugated molecular systems with chain-length dependent behavior

Yang-Yang Zhou[1], Yu-Chun Xu[1], Ze-Fan Yao [1], Jia-Ye Li[1], Chen-Kai Pan[1], Yang Lu[1], Chi-Yuan Yang[1], Li Ding[1], Bu-Fan Xiao[1], Xin-Yi Wang[1], Yu Shao [1], Wen-Bin Zhang [1], Jie-Yu Wang[1], Huan Wang[1] & Jian Pei [1] ✉

It remains challenging to understand the structural evolution of conjugated polymers from single chains to solvated aggregates and film microstructures, although it underpins the performance of optoelectrical devices fabricated via the mainstream solution processing method. With several ensemble visual measurements, here we unravel the morphological evolution process of a model system of isoindigo-based conjugated molecules, including the hidden molecular assembly pathways, the mesoscale network formation, and their unorthodox chain dependence. Short chains show rigid chain conformations forming discrete aggregates in solution, which further grow to form a highly ordered film that exhibits poor electrical performance. In contrast, long chains exhibit flexible chain conformations, creating interlinked aggregates networks in solution, which are directly imprinted into films, forming interconnective solid-state microstructure with excellent electrical performance. Visualizing multi-level assembly structures of conjugated molecules provides a deep understanding of the inheritance of assemblies from solution to solid-state, accelerating the optimization of device fabrication.

Multi-level assembly structures of conjugated polymers, ranging from single chain, solution-state aggregates to solid-state microstructures, have been proven to significantly influence the charge transport performance of polymeric electronic devices[1,2], such as solar cells[3–5], organic field-effect transistors (OFETs)[6–8], organic thermoelectrics (OTEs)[9–12], and so on. More recently, the relationship between molecular structures and solid-state microstructures has been intensively investigated[13–19]. Solution-state aggregation is a critical link connecting the molecular structure and the solid-state structure; however, its nature, the regulation methods, and its impact on the solid-state morphology remain elusive[20–23]. The lack of this knowledge hinders the understanding of the structure-property relationship of conjugated polymers and further the development of high-performance conjugated polymers and optoelectronic devices.

Effective characterization is critical in investigating the multi-level assembly structures of conjugated polymers. Nevertheless, multi-level assembly structures span broad spatial dimensions, ranging from Angstrom to micrometer, for which the characterization is challenging[24,25]. Recently, multiple methods have been applied to unravel these structures, including computational simulation, photophysical analysis, thermoanalysis, diffractions and scatterings, and so on[26–32]. For example, Yi and co-workers used molecular dynamics simulations to investigate the backbone conformation and aggregation behavior of conjugated polymers[33]. Panzer et al. meticulously analyzed the temperature-dependent absorption features of some typical conjugated polymers and demonstrated the order-disorder transition process of solvated aggregate structures[22]. Our previous work adopted small-angle neutron scattering to probe the mesostructures of polymer aggregates in different solvents[34]. Luo et al. used

[1]Beijing National Laboratory for Molecular Sciences (BNLMS), Key Laboratory of Polymer Chemistry and Physics of Ministry of Education, Center or Soft Matter Science and Engineering, College of Chemistry and Molecular Engineering, Peking University, Beijing 100871, China. ✉e-mail: jianpei@pku.edu.cn

ultrafast scanning calorimetry to unravel the ultrafast crystallization kinetics of conjugated polymer films[35]. These characterizations provide excellent circumstantial evidence. However, few experiments on direct observation of the multi-level assembly structures have been reported, which can provide information on the pathway, mechanism, and heterogeneity that are indistinguishable by ensemble measurements.

Herein, several advanced techniques were innovatively adopted to visualize an entire evolution venation of the multi-level assembly structures of conjugated polymers and study their structure-performance relationship. We presented the dynamic disaggregation process of conjugated polymers in organic solutions using liquid-phase electron microscopy (LP-EM). LP-EM has been expertly applied to imaging biological structures[36,37], electrochemical reactions[38–40], solution-phase nanoparticle growth[41], and so on[42]. Combined with other characterizations like conventional EM, atomic force microscopy (AFM) on freeze-dried samples, and in situ UV-*vis* absorption spectroscopy, we explored the multi-level assembly structures of isoindigo (IID)-based oligomers and polymers and studied their structure-performance relationship. We found that short polymer chains consistently show rigid conformation and form isolated solvated assemblies, which further grow to form solid-state microstructures with high crystallinity and low mobility during solvent drying. On the contrary, long polymer chains owing to their flexible conformation generate interconnected aggregates network in solution and directly form less crystalline interconnected microstructures with high mobility during drying. We show that, the conjugated polymers that are flexible and of high-molecular-weight can form conducive solid-state microstructures through multi-level assembly thereby improving the charge transport and significantly enhancing the electrical conductivity even in the doped state.

## Results

Monodispersed oligomers have well-defined molecular structures, thereby suitable for deducing the behavior of polymers as a function of chain length[43]. Thus, three IID-based oligomers of $m = 2, 3, 5$ and three polymers with different molecular weights were developed, named as (IID-DT)$_2$-IID, (IID-DT)$_3$-IID, (IID-DT)$_5$-IID, IIDDT$_{19}$, IIDDT$_{57}$, and IIDDT$_{99}$, respectively (Fig. 1a). The chemical structure and mono dispersity were confirmed by NMR spectroscopy (Fig. 1b, Methods, and Supplementary NMR Spectra). Long polymer chains with a broad dispersity were used as a blended system exhibiting homogeneous properties[44,45]. Soxhlet extraction and preparatory gel permeation chromatography (GPC) were used to purify and obtain smaller dispersity of the synthesized polymers (Fig. 1c and Supplementary Fig. 4). Table 1 summarized the corresponding properties of the samples including molecular weights, monomer numbers, dispersity, aggregates' sizes, and etc.

### Primary structure

Single chain conformation is an important parameter of the primary structure, which directly determines the aggregation behavior of polymer chains. To investigate how the chain length influences the chain conformation, cryogenic electron microscopy (cryo-EM) was performed on 0.1 g·L$^{-1}$ oligomer and polymer solutions (Fig. 1d and Supplementary Figs. 5–7). The samples of (IID-DT)$_2$-IID showed some spot-like features with diameter of 4–5 nm (marked with orange circles in Supplementary Fig. 5a), matching with the expected molecular size (Table 1). With the increase of chain length, we started to observe slender fibers. The widths of these fibers were around 3–4 nm, similar to the width of a single chain (~3.5 nm, Supplementary Fig. 8). Figure 1d shows the average lengths of these fibers counted from cryo-EM images and the comparison with calculated lengths of single chains for different molecular weights, the calculated data were nearly in the floating range of the statistical data. Therefore, these slender fibers

capture the single chain conformations. For short oligomers, (IID-DT)$_3$-IID, (IID-DT)$_5$-IID, and polymer IIDDT$_{19}$, the chains showed short rod-like conformation. While long chain polymers, IIDDT$_{57}$ showed richer coiled conformation than (IID-DT)$_3$-IID, (IID-DT)$_5$-IID, and IIDDT$_{19}$; The chains became more flexible with the increase of chain length: IIDDT$_{99}$ showed the most flexible chain conformation. The decreased stiffness with longer chain lengths impacts the chain-length dependent aggregation behavior of polymers.

### Dynamic disaggregation process

Despite their known importance in the microstructures of polymer films obtained from solution processing, few methods were able to visualize the dynamic disaggregation process directly. To do so, we trapped polymer solutions in graphene liquid pockets and imaged individual assembly structures with in situ LP-EM (Supplementary Note 1). Electron beam can induce bubbles from the radiolysis of solvents. Liquid bubble interface allows the observation of dynamic aggregation/disaggregation of solvated aggregates during the movement of bubbles, mimicking the gas-liquid interface during solution processing (Supplementary Note 2). The surface tension and shear force at the gas-liquid interface provide a sufficient driven force for the disaggregation of aggregates[42] (Supplementary Note 3). 1 g·L$^{-1}$ IIDDT$_{19}$ *o*DCB solution was used as the sample. Figure 2a shows the sample preparation to form graphene liquid pockets[46]. The liquid pockets were formed by laying one graphene sheet on top of a polymer solution laid graphene sheet that suspended on an EM grid. Pockets were typically rectangular-shaped, 100–200 nm wide, several hundred nanometers in length. We identified solvated aggregates from the size and morphology as the dark contrasted features, ~15-80 nm in length and 5–10 nm in width. Figure 2b shows representative images from the dynamic process (Supplementary Movie 1 and Fig. 9). Figure 2c exhibits the corresponding binarized images to help identify the image-tracking results (red marks). During the disaggregation, a thick fiber showed the tendency to split into two gracile threads, followed by dissociated into several shorter and thinner yarns. A similar process was also captured in repeated experiments (Supplementary Movie 2, Figs. 10, 11, and 12). Likewise, a similar disaggregation process (from a large aggregate to medium-sized aggregates and finally to dispersed small aggregates) was found for (IID-DT)$_5$-IID, which confirmed that polymer chains contain six donor-acceptor (D-A) units would form aggregation structure in solution (Supplementary Movie 3, Figs. 13, 14). The increase in the projected area and perimeter coincides with the splitting process of thick fiber, indicating the disaggregation process (Supplementary Figs. 9, 11, 14 and 15, and Supplementary Note 4). All analyzed samples remained intact under the used electron dose (Supplementary Fig. 16 and Supplementary Note 5). However, for longer chains, no solvated aggregates could be found in a single liquid pocket, likely because the sizes of aggregates, several hundred nanometers, were too large for the liquid pocket (typical width <300 nm). Unfortunately, such limitation prevents the comparisons across polymer chains of higher molecular weights, therefore we conducted ensemble absorption measurements and atomic force microscopy imaging to comprehensively evaluate the chain length effect on solution-state aggregation structures of these samples.

Temperature-dependent absorption spectroscopy was used to investigate the order-disorder transition of assemblies by stepwise increase of temperature (Fig. 2e–i and Supplementary Fig. 17)[22]. Ratios of absorption peaks unraveled the distinct aggregation behaviors of these samples. At room temperature, the ratio of the maximum absorption to its shoulder peak increased with the increase of chain length, indicating the enhanced aggregation tendency. The optical bandgap was related to the effective conjugation length of chains[47]. The spectra onsets for short oligomers, (IID-DT)$_2$-IID and (IID-DT)$_3$-IID, were constant at different temperatures (Fig. 2d, e and Supplementary Fig. 17). However, for (IID-DT)$_5$-IID, the variation of spectra onset

started to show a similar trend as polymers, indicating that a length around six D-A units was the critical length of chain conformation transition. The onset variation (Fig. 2d) further revealed three stages of the disaggregation process. (1) At low-temperature range, the optical bandgap was mainly determined by the aggregates. The effective conjugation length remained constant for different temperatures. (2) When it reached the critical temperature, the optical bandgap changed rapidly within a small temperature range. At this stage, the chains stayed at the threshold from aggregates to single chains, corresponding to the abrupt change of the effective conjugation length. The spectra variation was mainly due to the changing of backbone structures, from planar conformation in aggregates to torsional conformation in dispersed chains. (3) With the further increase of the temperature, we observed two distinctive features. The totally disaggregated chains as seen from the shorter and invariable effective

conjugation length, and the blue shift of the wide band at 550 nm–650 nm could be ascribed to the local torsion of molecular conformation. The three-stage disaggregation process analyzed from absorption spectra is also partially consistent with the imaging results of LP-EM. Besides, molecular dynamics simulations were also performed on $(IID-DT)_5$-IID to qualitatively analyze the disaggregated process. At 25 °C, the initial aggregate got swollen and showed a tendency to separate into two smaller aggregates, but the whole aggregate maintained long-range order. At 100 °C, the aggregate split into two 4-chain and 2-chain aggregates (Supplementary Figs. 18, 19). The tendency of the simulated results was consistent with our experimental observations[33,48]. Notably, as the chain length increased, the second stage was located at a higher temperature and showed a narrower range. From these results, we concluded that when the chain length exceeded a critical value (here was six D-A units), the chains

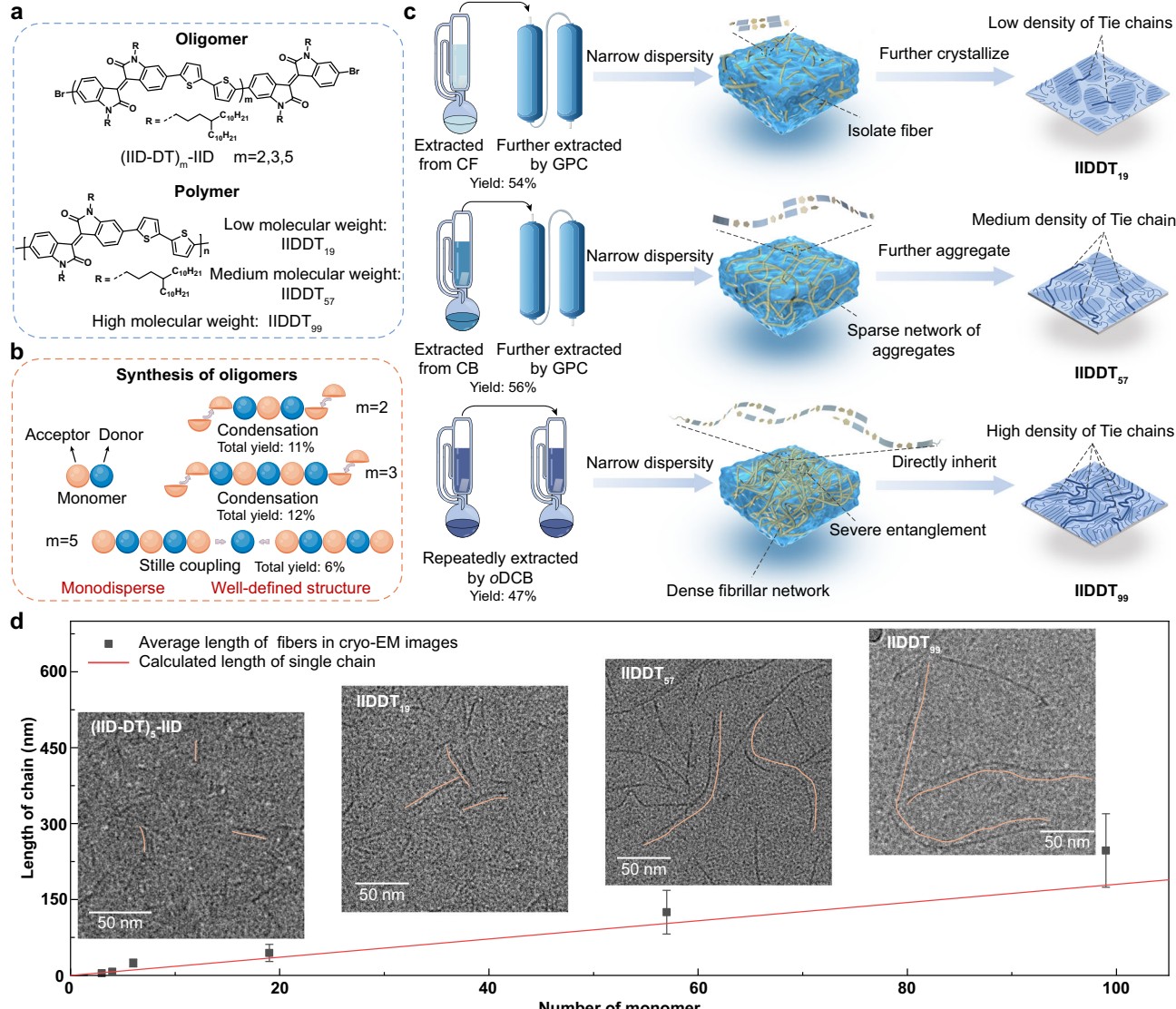

**Fig. 1 | Different chain conformations of IID-based oligomers and polymers.**
**a** Chemical structures of three IID-based oligomers and three polymers of different molecular weights. **b** Schematic of the synthesis process of three oligomers. **c** Schematics of the purification process of these three polymers and the different evolution of the corresponding multi-level assembly structures. CF: chloroform; CB: chlorobenzene; $o$DCB: $o$-dichlorobenzene. The schematics of film morphology (the right section) were adapted from ref. 6. **d** The average distribution of fiber length counted from cryo-EM images of $(IID-DT)_2$-IID, $(IID-DT)_3$-IID, $(IID-DT)_5$-IID, $IIDDT_{19}$, $IIDDT_{57}$, and $IIDDT_{99}$, their error bars were collected from 120 data, 90 data, 150 data, 154 data, 65 data, and 40 data, respectively. The red line shows the calculated length of a single chain with the number of monomers. Insets: representative cryo-EM images. The orange lines show the track of several fibers.

**Table 1 | Molecular weight and size of the assemblies of these samples**

| Polymer | Mn (kDa) | Average number of monomers | Dispersity | Calculated length of single chain(nm) | Average diameters of fibers in solutions (nm) | Average diameters of fibers in films (nm) |
|---|---|---|---|---|---|---|
| (IID-DT)₂-IID | 3.2 | 3 | 1 | 5.4 | / | / |
| (IID-DT)₃-IID | 4.4 | 4 | 1 | 7.2 | / | / |
| (IID-DT)₅-IID | 6.6 | 6 | 1 | 10.8 | 38.2 ± 8.9 | 78.4 ± 18.6 |
| IIDDT₁₉ | 20.7 | 19 | 1.4 | 34.2 | 40.8 ± 12.6 | 93.3 ± 35.4 |
| IIDDT₅₇ | 62.6 | 57 | 1.8 | 102.6 | 36.4 ± 14.2 | 36.8 ± 9.1 |
| IIDDT₉₉ | 108.9 | 99 | 2.8 | 178.2 | 21.4 ± 4.6 | 21.8 ± 4.7 |

The concentration of these solutions was 1 g·L⁻¹.

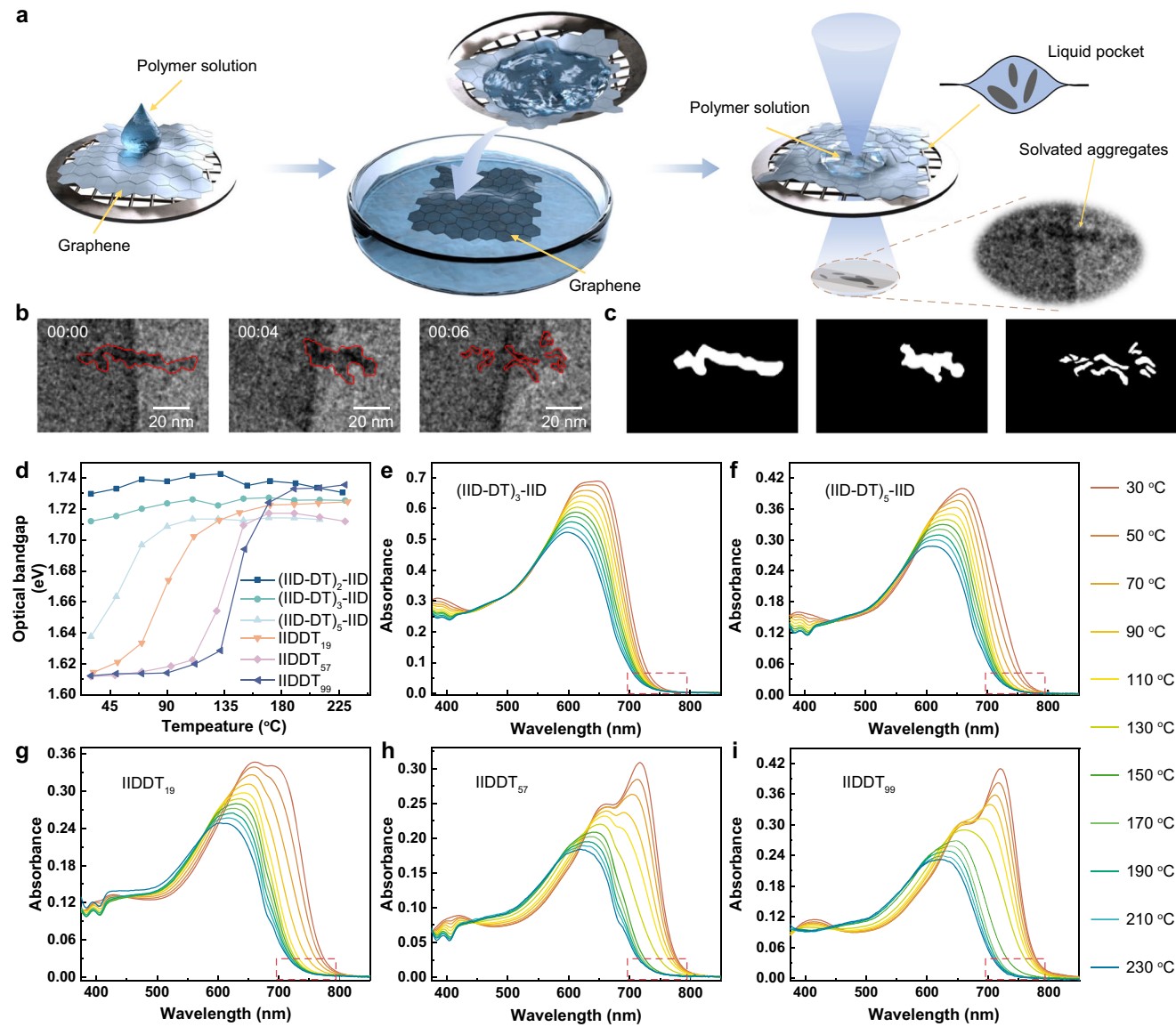

**Fig. 2 | Visualizing the dynamic disaggregation process of polymers in solutions. a** Schematic of sample preparation of graphene liquid pockets for in situ LP-EM study. **b** LP-EM images for the dynamic disaggregate process of IIDDT₁₉ in a liquid pocket, at the bubble/liquid interface, mimicking the condition for solution processing of conjugated polymer. The red marks identify the image-tracking results. Scale bar: 20 nm. The solution concentration is 1 g·L⁻¹. **c** The corresponding binarized images obtained from threshold intensity. **d** Temperature-dependent absorption spectra showing the optical bandgap variation as a function of temperature for the sample oligomers and polymers. **e–i** Temperature-dependent absorption spectra of 0.01 g·L⁻¹ oligomers and polymers in 1-chloronaphthalene (CN). The red dotted rectangles marked the onset variation regions of these spectra.

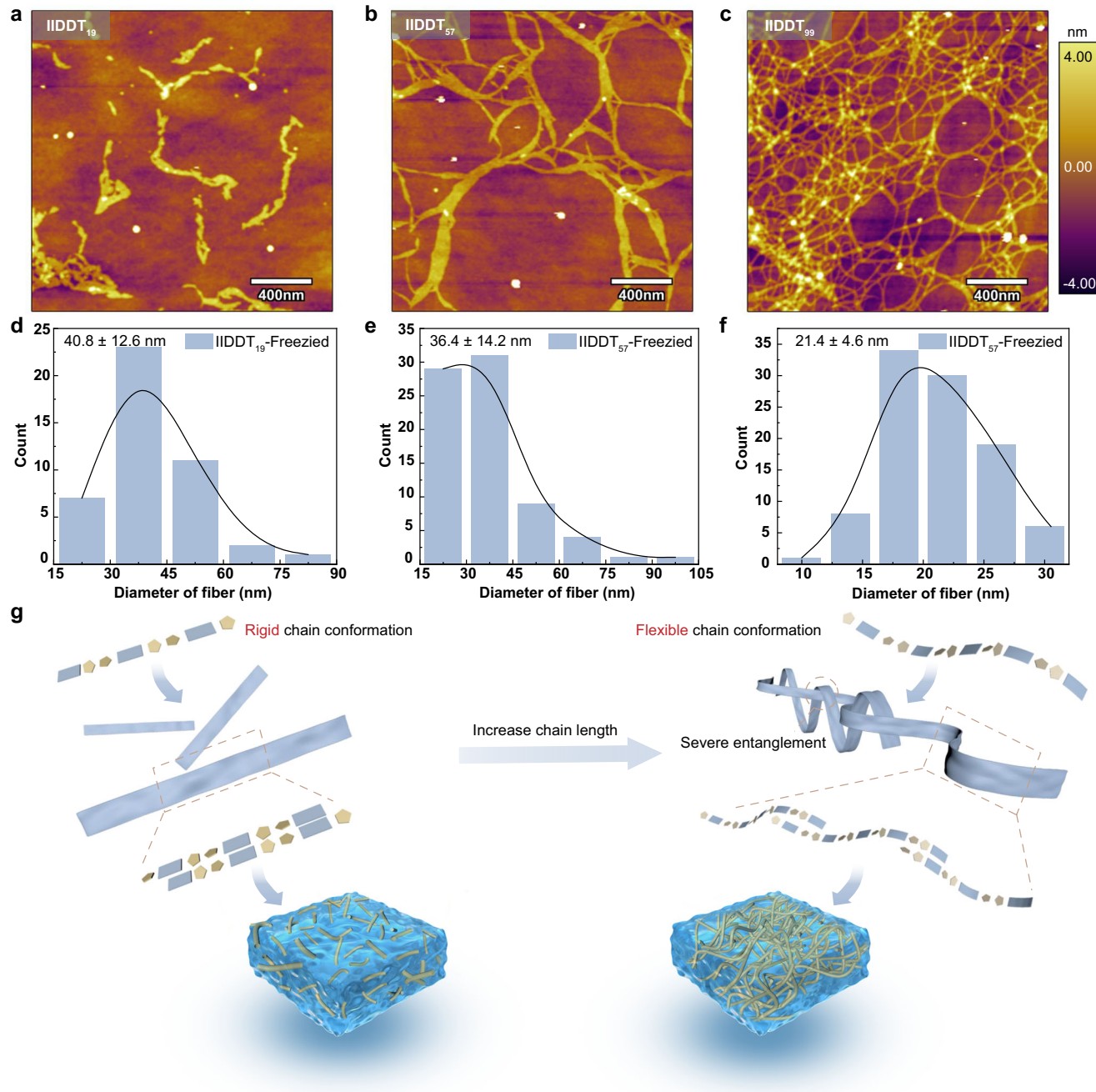

**Fig. 3 | Visualizing the assembly structures of polymers with different chain conformations. a–c** AFM height images of these polymers freeze-dried from $1\,g\cdot L^{-1}$ $o$DCB solutions. **d–f** Histograms of fiber diameters for IIDDT$_{19}$, IIDDT$_{57}$, and IIDDT$_{99}$, from the AFM height images of freeze-dried polymers. Sample sizes are 44, 75, and 98, respectively. **g** Proposed evolution of solution-state aggregates with chain-length dependent behavior.

would undergo a similar disaggregation process as the long chain but within different temperature windows.

## Structure of solution-state aggregates

To directly observe the solution-state aggregates of chains with different conformations, the sample solutions were dropped on Si substrates and freeze-dried (Fig. 3a–c and Supplementary Fig. 20). $1\,g\cdot L^{-1}$ was chosen as the appropriate concentration to avoid severe overlap of aggregates, such that clear aggregates could be found in the field of view. For (IID-DT)$_2$-IID and (IID-DT)$_3$-IID, the chains didn't form any aggregates in solutions. The sample of (IID-DT)$_5$-IID showed some dispersed fibers, with an average diameter of around $38.2 \pm 8.9\,nm$ (Table 1 and Supplementary Fig. 20). For IIDDT$_{19}$, isolated fibrillar aggregates were found randomly dispersed in solutions with an average fiber diameter of around $40.8 \pm 12.6\,nm$ (Fig. 3a, d and Table 1). For IIDDT$_{57}$, the medium chain length, the fibrillar aggregates were found to sparsely intersect with each other, producing a loosely crosslinked network of aggregates in solution. The average diameter of these fibers was around $36.4 \pm 14.2\,nm$ (Fig. 3b, e and Table 1). As for IIDDT$_{99}$, the longest polymer chains with flexible conformation, we found plentiful and slender fibers composed of a densely crosslinked network of aggregates, and the average diameter of these fibers was around $21.4 \pm 4.6\,nm$ (Fig. 3c, f and Table 1). In another word, short and rigid chains form isolated assemblies in solution (Fig. 3g). It is easy to

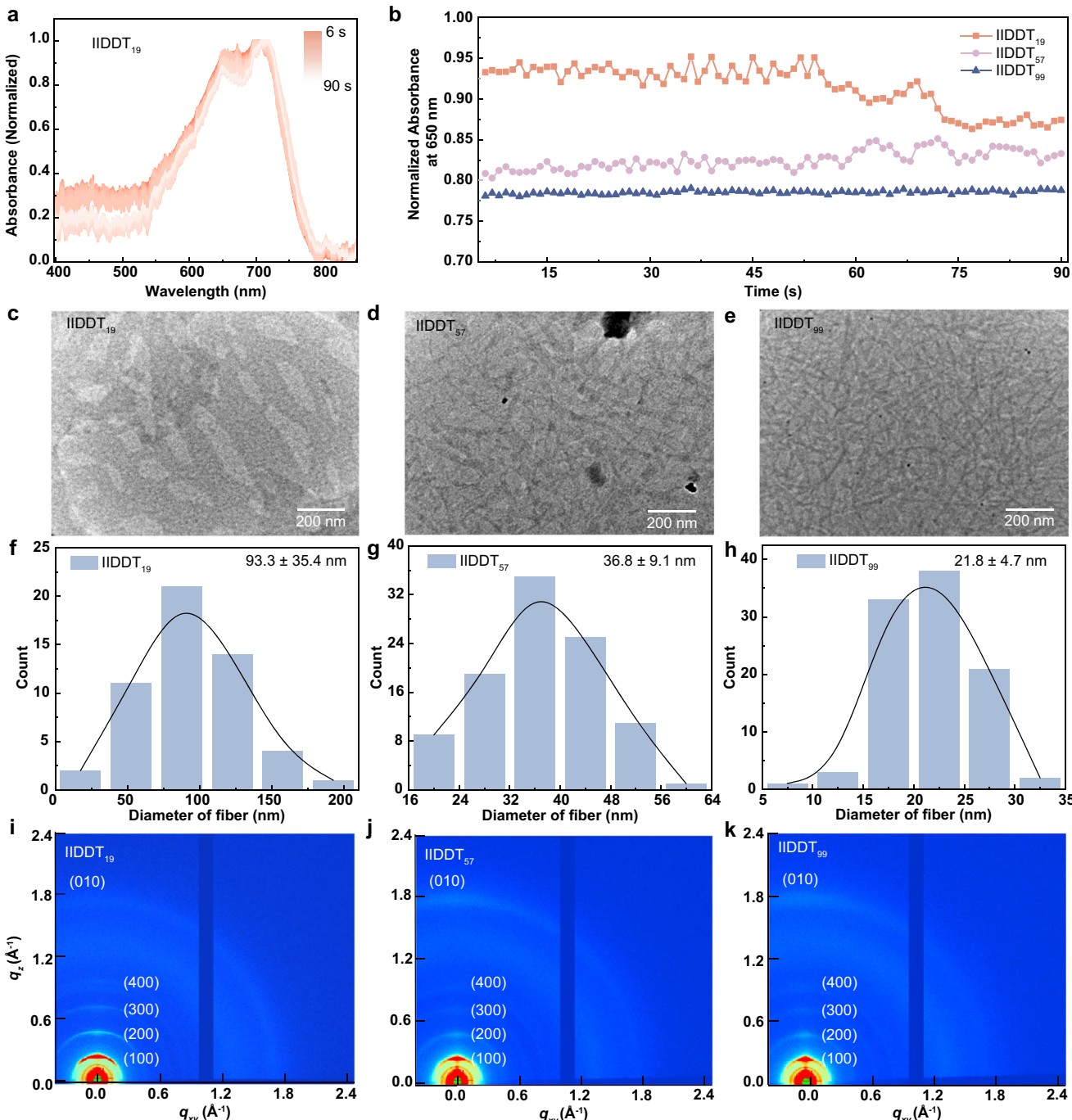

**Fig. 4 | Visualizing the evolution of aggregates during the deposition process of these polymers. a** Normalized UV-*vis* absorption spectra of the film formation process of IIDDT$_{19}$ as a function of time. **b** The normalized absorbance at 650 nm of these polymers during the in-situ film formation process. **c–e** EM images of IIDDT$_{19}$, IIDDT$_{57}$, and IIDDT$_{99}$ films deposited from 1 g·L$^{-1}$ *o*DCB solutions. **f–h** Histograms of fiber diameters for IIDDT$_{19}$, IIDDT$_{57}$, and IIDDT$_{99}$, which were counted from the EM height images of polymer films deposited from 1 g·L$^{-1}$ oDCB solutions, where 53, 100, and 98 datasets were collected, respectively. **i–k** GIWAXS patterns of IIDDT$_{19}$, IIDDT$_{57}$ and IIDDT$_{99}$ films spin-coated by 5 g·L$^{-1}$ *o*DCB solutions.

overcome the internal interaction of the isolated aggregates to make it totally disaggregate, corresponding to a low second temperature stage, as discussed above in absorption measurements. Nevertheless, for long chains, a single chain with flexible chain conformation may participate in multi-aggregates, and an interconnected network of aggregates eventually forms in solution (Fig. 3g). Thus, higher energy is necessary to disrupt the entire network of aggregates, corresponding to a higher second temperature stage, which is consistent with the above results (Fig. 2d)[49]. As described below, these distinct aggregate structures will further undergo chain-length dependent transition into

solid-state morphologies with distinct features inherited from solutions.

## Film formation process

The film formation process is a crucial step for solution-state aggregation to be inherited into the solid-state microstructure. Thus, in situ UV-*vis* absorption spectroscopy was performed to investigate the conformation and aggregation kinetics of these samples during the spin-coating process (Fig. 4a, b and Supplementary Figs. 21–23). Figure 4a shows the normalized absorption spectra of IIDDT$_{19}$ during

spin-coating for 60 s and then standing for 40 s. Other spectra of these samples are shown in Supplementary Fig. 21. For (IID-DT)$_2$-IID and (IID-DT)$_3$-IID, new characteristic absorptions at 685 and 694 nm appeared during the film formation process, respectively, indicating the transition from single chains to aggregates of the oligomer chains (Supplementary Fig. 21 and 23). For (IID-DT)$_5$-IID, the solution spectra started to exhibit similar characteristic absorption as those long chain polymers. The normalized absorption at 650 nm, revealing the different molecular conformation and the extent of chain aggregation, was plotted in Fig. 4b and Supplementary Fig. 23c[50,51]. The decreased absorption for (IID-DT)$_5$-IID and IIDDT$_{19}$ indicated that the chain conformations of short and rigid chains would further change after spin-coating. For IIDDT$_{57}$, the normalized absorption fluctuated from 0.80 to 0.84 during the spin-coating and standing process. However, once the film was annealed, the absorption decreased from 0.81 to 0.77, indicating that the polymer chains with medium length were flattened, from a coil-like to a pancake-like structure, during the annealing process (Supplementary Fig. 24). Moreover, for IIDDT$_{99}$, the normalized absorption kept unchanged either after spin-coating or annealing. Concluding from these results, we speculated that short and medium polymer chains were further planarized during the film formation process, but the long polymer chains mostly imprint the initial chain conformation from solution-state to solid-state.

Besides the in situ UV-*vis* absorption spectra, images were adopted to identify the differences in film morphology of these samples. 1 g·L$^{-1}$ *o*DCB solutions were dropped on copper girds and naturally evaporated to gain solid-state microstructures, and the EM images of these films are shown in Fig. 4c–e and Supplementary Fig. 25. (IID-DT)$_2$-IID and (IID-DT)$_3$-IID showed island-like microstructures in films, while (IID-DT)$_5$-IID started to exhibit a network-like morphology in solid-state microstructures, similar to long chain polymers. This further verified that the length of six D-A units was the critical value to exhibit polymer properties in this system. IIDDT$_{19}$ formed coarse and sparse fibers with an average diameter around 93.3 ± 35.4 nm (Fig. 4c, f and Table 1). There were significant changes from solution-state aggregates to solid-state microstructures of short polymer chains. IIDDT$_{57}$ showed a denser distribution of aggregate fibers whose average diameter was 36.8 ± 9.1 nm, showing statistically negligible difference with solvated aggregates (Fig. 4d, g and Table 1). Nevertheless, IIDDT$_{99}$ exhibited the densest network of aggregates with an average fiber diameter around 21.8 ± 4.7 nm, which is generally consistent with the average diameters of assembles in solution (Fig. 4e, g and Table 1). For short chains with rigid conformation, the aggregates further assembled from the solution-state structures. In contrast, the assemblies of long chains with flexible conformation are largely and directly inherited into solid-state microstructures. These results were also consistent with the observations on the in situ UV-*vis* absorption spectra.

## Solid-state microstructures

Beside ultra-thin films fabricated by 1 g·L$^{-1}$ solutions on copper grids, more uniform films were also spin-coated from 5 g·L$^{-1}$ concentrated *o*DCB solutions, which were also used for doping as follows. AFM was performed to investigate the microstructures of these films. Supplementary Fig. 26 shows the AFM height images of these oligomer and polymer films. (IID-DT)$_2$-IID and (IID-DT)$_3$-IID showed flake-like morphology with large roughness. (IID-DT)$_5$-IID began to exhibit some coarse fibers in the flake structures. For long-chain polymers, the films displayed smoother morphology of smaller roughness, and the fibers were thinner and denser as chain length increased, echoing the above results of EM images.

Grazing-incidence wide-angle X-ray scattering (GIWAXS) was performed to investigate the crystallinity of the dropped films (Fig. 4i–k and Supplementary Figs. 27, 28). (IID-DT)$_2$-IID and (IID-DT)$_3$-IID exhibited several specific diffraction peaks, from which the geometrical information of molecular packing could be obtained. Further studies on the well-defined structures of these molecules are currently in progress. With the increase of chain length, assemblies adopted increased lamellar distances and similar π-π stacking distances (Supplementary Table 1). Furthermore, coherence length ($L_c$) and paracrystallinity disorder (g factor) were used to determine the crystallinity of these polymer films quantitatively. Coherence length reflects the crystalline grain size, and paracrystallinity disorder shows an accumulation of intrinsic and extrinsic defects which yields a statistical static disorder in films[28]. A larger coherence length usually indicates higher crystallinity. As the chain length further increased, the (h00) diffraction peaks showed decreased coherence length and increscent paracrystallinity disorder (Supplementary Fig. 29). These results indicated that polymers with longer chains formed films of lower crystallinity. The distinct structures of solution-state aggregates yield the formation of different solid-state microstructures. On the basis of above results, we proposed an explanation for the distinct morphology resulted from different chain conformations (Supplementary Fig. 30). For short chains, the weak interchain interaction makes it easy to overcome the low crystallization energy barrier to produce large domains with high crystallinity. But for long chains, a single chain always participates in generating multiple aggregates, and the strong aggregation force and severe entanglement between long chains make it difficult to overcome the high energy barrier to disaggregate or disentangle to reform the morphology of high crystallinity. Thus, long chains with flexible single chain conformation tend to directly inherit the solution-state aggregation structures and retain the high density of tie chains within the network of aggregates.

## Electrical performance

Since the distinct solid-state microstructures were inherited from the solution-state aggregations, it was expected that these samples would exhibit different charge-transport properties. Thus, polymer FET devices with top-contact/bottom-gate (TC/BG) configurations were fabricated using these materials as the active layers. With the increased chain length, the mobility increased from (7.28 ± 1.1) × 10$^{-4}$ cm$^2$ V$^{-1}$ s$^{-1}$ (for (IID-DT)$_2$-IID) to 2.55 ± 0.4 cm$^2$ V$^{-1}$ s$^{-1}$ (for IIDDT$_{99}$) (Fig. 5a and Supplementary Fig. 31). The high mobility of IIDDT$_{99}$ mainly benefits from the interconnective fibrillar aggregates with high-density tie chains, which were directly inherited from solution. Besides, the electrical performance could also be evaluated by measuring the conductivity of the doped samples. FeCl$_3$ was chosen as the dopant because of its outstanding doping ability for many p-type polymers[52,53]. The microstructures of these thin films were greatly retained after doping (Supplementary Fig. 32), while the packing order was slightly disrupted (Supplementary Figs. 33–35). The optimized electrical conductivities of these samples with different chain conformations are shown in Fig. 5b. The conductivities showed a positive correlation with chain length. Ultrashort oligomer (IID-DT)$_2$-IID exhibited the lowest conductivity of (9 ± 2) × 10$^{-3}$ S cm$^{-1}$. In contrast, polymer IIDDT$_{99}$ with extended and flexible chain conformation, showed the highest conductivity of 913 ± 35 S cm$^{-1}$ with the maximum value of 957 S cm$^{-1}$, almost five orders of magnitude higher than (IID-DT)$_2$-IID. It must be mentioned that the charge transport properties of conjugated molecular system vary by several orders of magnitude with molecular weight, suggesting that it is very important to study and regulate molecular weight of conjugated polymers. Notably, the maximum conductivity of IIDDT$_{99}$ also approached the conductivity of the commercially available conducting polymer PEDOT:PSS[54]. Figure 5c and Supplementary Figure 36 show the change of conductivity of these samples with doping time. At the same doping concentration, all these samples rapidly reached saturated conductivity, indicating that all of them exhibited high reactivity with FeCl$_3$.

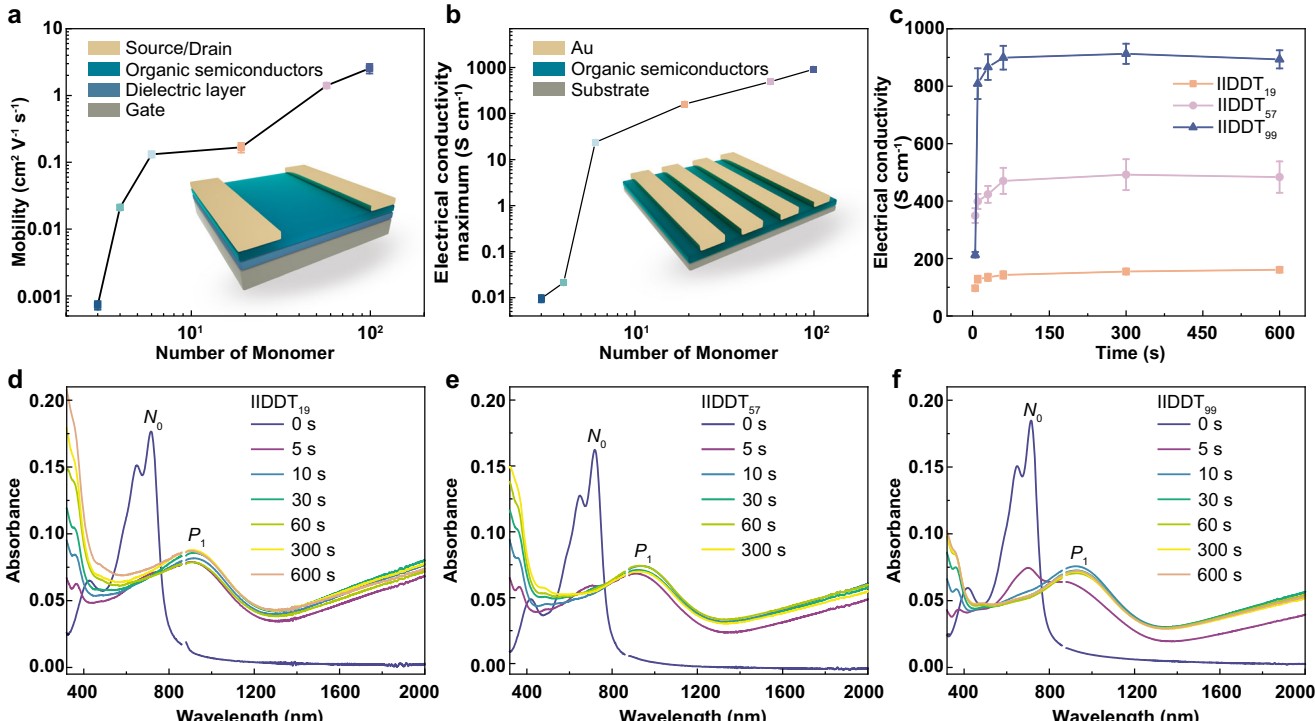

**Fig. 5 | Electrical performance of oligomers and polymers. a** The dependence of mobility on the chain length. Inset: Diagram of a polymer transistor. Due to the limitation of solution viscosity, the FET devices of $(IID-DT)_2$-IID and $(IID-DT)_3$-IID were fabricated with 20 mg mL$^{-1}$ trichloro ethylene solutions. The error bar of each molecule was collected from more than 10 devices. **b** Maximal electrical conductivity of the samples doped by $FeCl_3$. The error bar of each molecular was collected from more than 5 devices. **c** Electrical conductivities of these polymers doped for 5, 10, 30, 60, 300, and 600 s, respectively. Each error bar was collected from more than 5 devices. **d–f** UV-*vis*-NIR absorption spectra of these polymer films plotted as a function of time (0, 5, 10, 30, 60, 300, and 600 s) with doping (The break between 860 nm and 880 nm was ascribed to the fluctuation of the instrument.).

UV/visible/near-infrared (UV-*vis*-NIR) absorption spectroscopy was performed to investigate the doping level of these samples (Fig. 5d–f and Supplementary Fig. 37). Upon doping, the absorption band of the neutral samples (500–800 nm) diminished, while new characteristic absorptions at the NIR region, at around 900 nm ($P_1$) and >1300 nm ($P_2$), appeared due to the polaron or bipolaron generation. Besides, these samples showed distinct absorption at 322 nm and 364 nm after doping, which could be ascribed to the characteristic absorption of $FeCl_4^-$ and residual $FeCl_3$[55]. Moreover, the specific doping efficiency was quantified from the ratios of $A_{P1}/A_{N0}$ ($A_{P1}$: the absorbance of $P_1$ peak; $A_{N0}$: the intensity of neutral peak in pristine film) (Supplementary Fig. 38). For $(IID-DT)_2$-IID and $(IID-DT)_3$-IID, the high doping efficiency might be ascribed to the coarse morphology of film, which created lots of cracks for dopants to permeate and react with molecules. $(IID-DT)_5$-IID and these three polymers exhibited similar doping level when the films were sufficiently doped, indicating that all of them generated comparable polarons or bipolarons. The slightly lower doping level of $IIDDT_{99}$ might be attributed to the fact that the uniform and compact morphology with fewer grain boundaries prevented excessive dopants from penetrating into the film. On the basis of these results, we concluded that the high charge transfer performance (including the carrier mobility and electrical conductivity) of $IIDDT_{99}$ benefited from its interconnected microstructure with high density of tie chains, which was almost directly inherited from solution.

## Discussion

In summary, we have systematically investigated the effect of chain length on solution-state aggregation and film formation kinetics for conjugated systems. Notably, we have demonstrated that the high crystallinity of short chains with rigid conformation might mainly originate from the ordered growth of aggregates during film formation. However, the grain boundaries limited the charge transport properties. By contrast, for polymers with long and flexible backbones, the interconnected microstructure of polymer film almost directly originated from solution-state aggregates. That was, plenty of flexible chains connected multiple fibrillar aggregates to form assembled networks in solution, and after the direct inheritance, these flexible chains would serve as high-density tie chains to construct an effective charge transport network in films. Hence, $IIDDT_{99}$, the polymer with long and flexible chain conformation, exhibited the highest carrier mobility up to 3.23 cm$^2$ V$^{-1}$ s$^{-1}$ on neutral state and the highest conductivity with a maximum value of 957 S cm$^{-1}$ after doped. Our work reports the direct visualization of structural evolution of isoindigo-based conjugated molecular aggregates in solution, which filled the gap in the effect of chain-length dependent aggregation on the charge transport properties of D-A conjugated molecular system. It should be pointed out that the charge transport properties of conjugated molecular system vary by several orders of magnitude with molecular weight, thus it is very important to investigate and regulate chain length of conjugated polymers. Compared with other methods to tune solution-state aggregation of conjugated polymers, the modulation of chain length reported here provides a general strategy to optimize the performance of conjugated-polymer-based electronics.

## Methods

### Materials

All commercially available chemicals were used without further purification unless otherwise noted (Supplementary Table 2). The entire synthesis procedures of these oligomers are shown in Supplementary Information. The crucial synthesis procedures of these oligomers and polymers are shown below.

## (IID-DT)$_2$-IID

To a solution of (E)-6,6′-((1,1′-bis(4-decyltetradecyl)-2,2′-dioxo-[3,3′-biindolinylidene]-6,6′-diyl)bis([2,2′-bithiophene]-5,5-diyl))bis(1-(4-decyltetradecyl)indoline-2,3-dione) (250 mg, 0.112 mmol) and TsOH·H$_2$O (21 mg, 0.112 mmol) in toluene (20 mL) and HOAc (6 mL), a solution of 6-bromo-1-(4-decyltetradecyl)indolin-2-one (135 mg, 0.247 mmol) in toluene (10 mL) was added. After 24 h at 100 °C and then the mixture was allowed to warm to room temperature. After quenched with water, the mixture was extracted with chloroform, the organic layer was dried over anhydrous Na$_2$SO$_4$. After the solvent was removed under reduced pressure, the residue was purified by column chromatography on silica gel with eluent (PE:DCM = 2:1) to give crude product. The crude product was purified with preparative GPC with CHCl$_3$ as the eluent to give (IID-DT)$_2$-IID as black solids (195 mg, 53%). $^1$H NMR (400 MHz, CD$_2$Cl$_2$, ppm): δ 8.92–8.88 (m, 6H), 7.03–7.01 (m, 4H), 6.89–6.80 (m, 6H), 6.75 (m, 4H), 6.64 (m, 4H), 6.43–6.40 (m, 4H), 3.67–3.58 (m, 12H), 1.71–1.59 (m, 12H), 1.44–1.25 (m, 234H), 0.87–0.83 (m, 36H). $^{13}$C NMR (101 MHz, CDCl$_3$) δ 168.0, 167.9, 167.6, 162.5, 153.8, 149.3, 145.3, 145.2, 144.8, 142.9, 142.5, 137.9, 137.4, 137.1, 136.4, 132.6, 131.0, 130.7, 125.9, 124.6, 121.1, 120.8, 120.6, 110.9, 103.9, 103.5, 97.1, 37.2, 33.6, 32.0, 31.0, 30.3, 30.2, 29.8, 29.8, 29.7, 29.4, 26.7, 22.7, 14.2. MALDI-HRMS calcd. for C$_{208}$H$_{321}$Br$_2$N$_6$O$_6$S$_4$ ([M + H]$^+$): 3285.2242; Found: 3285.2244.

## (IID-DT)$_3$-IID

To a solution of (3E,3″E)-6′,6‴-([2,2′-bithiophene]-5,5′-diyl)bis(1,1′-bis(4-decyltetradecyl)-6-(5′-(1-(4-decyltetradecyl)-2,3-dioxoindolin-6-yl)-[2,2′-bithiophen]-5-yl)-[3,3′-biindolinylidene]-2,2′-dione) (74 mg, 0.020 mmol) and TsOH·H$_2$O (6.8 mg, 0.017 mmol) in toluene (10 mL) and HOAc (5 mL), a solution of 6-bromo-1-(4-decyltetradecyl)indolin-2-one (28 mg, 0.051 mmol) in toluene (10 mL) was added. After 24 h at 100 °C and then the mixture was allowed to warm to room temperature. After quenched with water, the mixture was extracted with chloroform, the organic layer was dried over anhydrous Na$_2$SO$_4$. After the solvent was removed under reduced pressure, the residue was purified by column chromatography on silica gel with eluent (PE:DCM = 1:1) to give crude product. The crude product was purified with preparative GPC with CHCl$_3$ as the eluent to give (IID-DT)$_3$-IID as black solids (38 mg, 39%). $^1$H NMR (500 MHz, C$_2$Cl$_4$D$_2$, 90 °C, ppm): δ 9.24-9.23 (m, 6H), 9.13-9.11 (d, J = 8.6 Hz, 2H), 7.37-7.36 (m, 6H), 7.30-7.27 (m, 6H), 7.24-7.23 (m, 6H), 7.19-7.18 (dd, J = 8.6 Hz, J = 1.6 Hz, 2H), 6.96 (m, 6H), 6.94 (m, 2H), 3.87-3.74 (m, 16H), 1.82-1.74 (m, 16H), 1.43-1.29 (m, 312H), 0.93-0.89 (m, 48H). $^{13}$C NMR (125 MHz, C$_2$Cl$_4$D$_2$, 90 °C, ppm): δ 168.0, 167.5, 145.8, 145.3, 143.1, 142.8, 138.3, 137.6, 137.0, 133.6, 132.9, 131.7, 131.5, 131.1, 131.0, 130.5, 129.9, 128.9, 126.0, 125.0, 124.6, 123.2, 121.5, 120.7, 120.3, 119.2, 118.8, 112.1, 111.0, 104.1, 103.9, 40.5, 37.8, 34.2, 31.7, 31.2, 30.0, 29.5, 29.1, 26.7, 24.7, 22.4, 13.8. MALDI-HRMS calcd. for C$_{280}$H$_{429}$Br$_2$N$_8$O$_8$S$_6$ ([M + H]$^+$): 4382.0094; Found: 4381.9937.

## (IID-DT)$_5$-IID

(IID-DT)$_2$-IID (167 mg, 0.051 mmol), 2,5-bis (trimethylstannyl)thiophene (5.86 mg, 0.012 mmol), Pd$_2$(dba)$_3$ (0.46 mg, 4 mol%), P(o-tol)$_3$ (0.61 mg, 16 mol%) and 10 mL of toluene were added to a Schlenk flask. The flask was charged with N$_2$ through a freeze-pump-thaw cycle for three times. After 12 h at 110 °C and then the mixture was allowed to warm to room temperature. The crude product was purified with preparative GPC with CHCl$_3$ as the eluent to give (IID-DT)$_5$-IID as black solids (45 mg, 57%). $^1$H NMR (500 MHz, C$_2$Cl$_4$D$_2$, 90 °C, ppm): δ 9.24–9.09 (m, 12H), 7.40-6.84 (m, 50H), 3.86–3.77 (m, 24H), 1.81–1.78 (m, 24H), 1.31 (m, 468H), 0.93–0.91 (m, 72H). MALDI-HRMS calcd. for C$_{424}$H$_{645}$Br$_2$N$_{12}$O$_{12}$S$_{10}$ ([M + H]$^+$): 6575.5798; Found: 6575.5775.

## IIDDT$_{19}$

6,6′-dibromo-N,N′-(4-decyltetradecyl)-isoindigo (80 mg, 0.073 mmol), 5,5′-bis(trimethylstannyl)-2,2′-bithiophene (43.19 mg, 0.088 mmol), Pd$_2$(dba)$_3$ (2.67 mg, 4 mol%), P(o-tol)$_3$ (3.54 mg, 16 mol%), and 7 mL of toluene were added to a Schlenk tube. The tube was charged with N$_2$ through a freeze-pump-thaw cycle for three times. The mixture was stirred for 0.5 h at 110 °C. N,N′-Diethylphenylazothioformamide (10 mg) was then added and then the mixture was stirred for 0.5 h to remove any residual catalyst before being precipitated into methanol (200 mL). The precipitate was filtered through a nylon filter and purified via Soxhlet extraction for 8 h with acetone, 12 h with hexane, and finally was collected with chloroform. The component extracted from chloroform was then further purified by preparatory gel permeation chromatography (GPC). The purified solution was then concentrated by evaporation and precipitated into methanol (200 mL) and filtered off to afford black solids (50 mg, 62%).

## IIDDT$_{57}$

6,6′-dibromo-N,N′-(4-decyltetradecyl)-isoindigo (80 mg, 0.073 mmol), 5,5′-bis(trimethylstannyl)-2,2′-bithiophene (35.99 mg, 0.073 mmol), Pd$_2$(dba)$_3$ (2.67 mg, 4 mol%), P(o-tol)$_3$ (3.54 mg, 16 mol%), and 7 mL of toluene were added to a Schlenk tube. The tube was charged with N$_2$ through a freeze-pump-thaw cycle for three times. The mixture was stirred for 1.5 h at 110 °C. N,N′-Diethylphenylazothioformamide (10 mg) was then added and then the mixture was stirred for 0.5 h to remove any residual catalyst before being precipitated into methanol (200 mL). The precipitate was filtered through a nylon filter and purified via Soxhlet extraction for 8 h with acetone, 12 h with hexane, 12 h with chloroform, and finally was collected with chlorobenzene. The component extracted from chlorobenzene was then further purified by preparatory GPC. The purified solution was then concentrated by evaporation and precipitated into methanol (200 mL) and filtered off to afford black solids (36 mg, 45%).

## IIDDT$_{99}$

6,6′-dibromo-N,N′-(4-decyltetradecyl)-isoindigo (80 mg, 0.073 mmol), 5,5′-bis(trimethylstannyl)-2,2′-bithiophene (35.99 mg, 0.073 mmol), Pd$_2$(dba)$_3$ (2.67 mg, 4 mol%), P(o-tol)$_3$ (3.54 mg, 16 mol%), and 7 mL of toluene were added to a Schlenk tube. The tube was charged with N$_2$ through a freeze-pump-thaw cycle for three times. The mixture was stirred for 6 h at 110 °C. N,N′-Diethylphenylazothioformamide (10 mg) was then added and then the mixture was stirred for 0.5 h to remove any residual catalyst before being precipitated into methanol (200 mL). The precipitate was filtered through a nylon filter and purified via Soxhlet extraction for 8 h with acetone, 12 h with hexane, 12 h with chloroform, 12 h with chlorobenzene, and finally was repeatedly extracted with o-dichlorobenzene. The purified solution was then concentrated by evaporation and precipitated into methanol (200 mL) and filtered off to afford black solids (28 mg, 35%).

## Experimental details

Molecular weights were determined by GPC performed on Polymer Laboratories PL-GPC220 at 150 °C using 1,2,4-trichlorobenzene (TCB) as eluent. Temperature-dependent absorption spectra were recorded on PerkinElmer Lambda 750 UV-vis spectrometer. LP-EM experiments were performed on a JEM2100 instrument with a Gatan Oneview IS camera, at the Analytical Instrumentation Center of Peking University, and a Tecnai T20 instrument, located at the Electron Microscopy Laboratory of Peking University, both at 80 kV. Conventional transmission electron microscopy (EM) images were obtained by FEI Tecnai T20 operated at accelerating voltage of 200 kV. The grazing incidence wide-angle X-ray scattering (GIWAXS) was recorded with a Ganesha SAXS instrument. In GIWAXS tests, a monochromated X-ray radiation source with a wavelength λ of 0.154 nm was used, and then the detection was performed with a hybrid photon counting (HPC, Dectris, PILATUS3 R, 300 K) detector. The distance from the sample to the detector was 83.44 mm. The grazing angle was 0.2°. Coherence length was calculated from Scherrer equation by simulating the full width at

half maximum of (100) diffraction peaks of polymer thin films. Para-crystalline disorder (g factor) was calculated from the center position ($Q_O$) and breadth ($\Delta_Q$) of a diffraction peak: $g^2 = \Delta_Q/(2\pi Q_0)$. Atomic force microscopy studies of thin films were performed with Cypher S microscope (Asylum Research, Oxford instruments) at tapping mode under ambient conditions using silicon cantilever (AC240TS-R3) with a resonant frequency around 70 kHz. The in situ UV-*vis* absorption spectrum were studied using a home-made in situ absorption device.

### Liquid-phase electron microscope (LP-EM)

We show the procedure of making the liquid pocket in Fig. 2a. First, we putted gold grid with mesh size 300 on copper which was grown single layer graphene (ACS Materials). Then, we used 0.1 M ammonium per-sulphate as etchant to etch the residual copper and graphene-covered gold EM grid (SPI supplies) was obtained. Next, 0.5 μL polymer solution was dropped on the graphene-covered gold EM grid, and the grid was turned and gently placed on freely-floating 3-5 layers graphene. After waiting for several minutes, we removed the grid and put it on clean filter paper for residual water evaporating. Besides, for Supplementary Movie 1, the dose rate of each frame was 14.5 e⁻·Å⁻²·s⁻¹; for Supplementary Movie 2, the dose rate was 9.6 e⁻·Å⁻²·s⁻¹; for Supplementary Movie 3, the dose rate was 3.9 e⁻·Å⁻²·s⁻¹. These electron doses were within the typical ranges used for organic materials, under which structures remain intact[46,56,57].

### Cryogenic electron microscopy (Cryo-EM)

For sample preparation, 1.5 μL solution was dropped on a copper grid, and the excess solution was blotted for 3 s by a filter paper. Then, the grid was immediately plunged into liquid ethane with an FEI Vitrobot Mark IV (25 °C and 0% humidity). Next, the grid was plunged into liquid nitrogen for freezing the solution. Cryo-grids of samples were screened in a FEI Titan Krios operated at 300 kV with a Gatan K2 (GIF) direct electron detector for images collection. The images were collected at a nominal magnification of x130000. The dose rate of each image was about 7 e⁻·Å⁻²·s⁻¹ with a total exposure time of 4 s.

### Thin film fabrication and immersion doping

All devices were fabricated through spin-coating polymer solutions on substrates. The substrates were subjected to cleaning using ultra-sonication in acetone, deionized water (twice), and isopropyl alcohol. Thin films were deposited on the substrates by spin-coating 5 g·L⁻¹ polymer solutions at 1000 rpm for 60 s and 2000 rpm for 3 s, and followed by annealing at 180 °C for 10 min. The preparation of the samples for GIWAXS, AFM and SEM analysis uniformly used Si substrates. All doping experiments were carried out in air atmosphere. 20 mM FeCl₃ solution was prepared by dissolving 97 mg of FeCl₃ in 30 mL anhydrous nitromethane in a glass vial and then were divided into six equal solutions. Doping was done by dipping the spin-coated polymer film in the dopant solution for several seconds, and then the residual solution was blown away by ear washing bulb.

### FET device fabrication and characterization

For polymer thin-film transistors, top-contact/bottom-gate (TC/BG) device configuration was adopted. The FET devices were fabricated using n⁺⁺-Si/SiO₂ (300 nm) substrates, and the substrates were modified with octadecyltrimethoxysilane (OTS) to form a SAM monolayer. 5 g·L⁻¹ polymer solutions were spin-coated on the OTS modified substrates at 1000 rpm for 60 s and 2000 rpm for 3 s. After annealing at 180 °C for 10 min, 40 nm Au was deposited under vacuum ($4 \times 10^{-4}$ Pa) via shadow mask method as the top source and drain electrodes. The evaluations of the transistors were carried out in the atmosphere (RH = 30 ~ 40%) on a probe stage using a Keithley 4200 SCS as the parameter analyzer. The FET devices had a channel length ($L$) of 30 μm and a channel width ($W$) of 1200 μm. The carrier mobilities, $\mu$, were calculated from the data in the saturates regime according to the equation $I_{DS} = (W/2L)C_i\mu(V_G\text{-}V_T)^2$,

where $I_{DS}$ is the drain current in the saturated regime, $C_i$ (3.7 nF·cm⁻² for 500 nm Cytop) is the capacitance per unit area of the gate dielectric layer, $V_G$ and $V_T$ are the gate voltage and threshold voltage. $V_T$ of the device was determined from the relationship between the square root of $I_{DS}$ and $V_G$ at the saturated regime.

### Molecular dynamics simulations

All-atom molecular dynamics simulations were performed using Gromacs-2019.4 with GPU acceleration[58]. General AMBER force field (GAFF) was adopted to descript the interactions between atoms[59]. Atomic charges were calculated from the restrained electrostatic potentials (RESP) at the ωB97XD/6-31 G(d) level of theory using Multiwfn program[60,61]. The initial models for the molecular dynamics simulations were built by constructing oligomer stacks (consisting of six oligomer chains perfectly packed on top of each other) (Supplementary Fig. 18); each oligomer chain is 6 repeat-units long. The simulation boxes consisted of 102708 atoms in total, including 8000 *o*DCB molecules. The molecular dynamics simulations were performed at 25 °C (for 20 ns) or 100 °C (for 20 ns).

## Data availability

The source data generated in this study have been deposited in Figshare (https://doi.org/10.6084/m9.figshare.22651477) and are also available from the corresponding author upon request.

## Code availability

The codes that support the findings of this study are available from the corresponding author upon request.

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

## Acknowledgements
This work is supported by National Key R&D Program of China (2017YFA0204701), National Natural Science Foundation of China (21790360, 21722201, 21925102, 22174006). Beijing Outstanding Young Scientist Program (BJJWZYJH01201910001001). The authors thank the Cryo-EM Core Facility Platform and Laboratory of Electron Microscopy at Peking University. The authors thank Prof. Huanping Zhou and Dr. Liang Li for their help in the measurement of in situ UV-*vis* absorption spectrum.

## Author contributions
Y.-Y.Z. and B.-F.X. synthesized and purified these polymers. Y.-C.X. synthesized the oligomers. C.-K.P. performed the molecular dynamics simulations. J.-Y.L. and Y.-Y.Z. performed the LP-EM experiments. Y.-Y.Z. performed the temperature-dependent absorption spectra. Y.-Y.Z. and Y.-C.X. performed the freeze-dried experiments. Y.-Y.Z. and L.D. measured and analyzed the EM images of polymer films. Y.-Y.Z. and X.-Y.W. measured and analyzed the GIWAXS and AFM data. Y.-Y.Z. fabricated and measured the OFET devices. Y.-Y.Z. measured the conductivities of these oligomers and polymers. Y.-Y.Z., Z.-F.Y., Y.L., C.-Y.Y., Y.S., W.-B.Z., J.-Y.W., H.W., and J.P. wrote the manuscript.

## Competing interests
The authors declare no competing interests.
