## [Peer Review File · Nature Communications]

Visualizing the multi-level assembly structures of conjugated molecular systems with chain-length dependent behaviorReviewers' Comments:

Reviewer #1:

Remarks to the Author:

This manuscript by Zhou, Y.-Y et al. demonstrates an analysis on assembled polymeric structures that depend on chain lengths of monomers, and relates the structures to functionality. The authors have performed experiments in multiple scales using various techniques including cryo-EM, graphene liquid-phase TEM, UV-vis, X-ray scattering, and charge transport measurements. Although the study is interesting and may be beneficial for understanding structural effects on performance for polymeric devices, the conclusions made in this study have already been established previously in other publications and do not seem to bring enough new insights in its current form. However, having following concerns fully addressed, the work can be recommended for a publication in Nature Comm.

1. One suggestion may be to provide further quantitative analysis and solid conclusions on the results for the solution-phase observations of polymeric chains using TEM, which could bring about information into the dynamics that may govern the assembly or disassembly process. Some questions I had were: (1) Is there a relationship between the location of the bubble interface within the graphene liquid cell and the location of the polymeric disassembly onset? (2) How significant is the gas-liquid interface in actual solution processes, and how well can the graphene liquid cell, which is a confined environment, model the actual disassembly process that occurs in batch solution? The bubbles formed are radiolysis products, and will also have a different gas composition. (3) Are there changes in the dynamics of the separated polymer strands once disassembly occurs that could affect the overall assembly process?
2. The authors should additionally provide information on the beam dose rate and the threshold electron dose of the polymers, and comment on how tolerant the polymers are to the electron beam.
3. Maybe the authors could use other liquid cell designs to capture the dissolution process of the longer polymers, as there are a variety of liquid cells available. Using a spacer within the graphene sandwiches may allow a bigger sized liquid pocket.
4. What is the gray shaded area in figures 2(e-i)? We noticed that there is also a gray 'shadow' next to the legend in figure 2. Also, perhaps reorganize figures 2(e-i) so that they are separated from figure 2(d), as the color scheme of the two sets of figures are similar (while one is for polymer type and one is for different temperatures) and may cause confusion.
5. Additionally, there were several grammatical errors that must be corrected in the manuscript. Some I highlight are:
 - a. "In the contrast..." (Line 67) should be "In contrast" or "On the contrary".
 - b. "longer polymer chains owing to its flexible conformation..." (Line 68) should be "to their" instead of "its"
 - c. "long polymer chains with a broad dispersity were usually used..." (line 79) should be "were used" if you used it in this experiment.

Reviewer #2:

Remarks to the Author:

Charge transport behaviors of conjugated polymers are complex, arising from the interplay of structural and morphological influence of polymer chains. Due to the polydispersity nature of polymer chains, it is rather difficult to disentangle this influence. This manuscript is beautifully carried out and attempts to reveal charge transport behaviors by revealing the assembly structures. This is made possible by carefully fractioning polymer chains. What is appealing in this study is the virtualization of chain-length dependent assembly behaviors. Many of the observations have been known and

demonstrated, such as crystalline structures from short chains and tie-chain connections from long chains. In a previous study, Chaudhry et al even made well-defined discrete oligomers to illustrate chain dynamics. Salleo, Diao, and many others have done nice work to help understand assembly behaviors. This study is kinda like the Jewel in the Crown and elegantly pieces important aspects together to provide a whole picture. In my opinion, it will be well received in the community.

There is confusion in the conclusion "Hence, IIDDT99, the polymer with long and flexible chain conformation, exhibited the highest carrier mobility up to $3.23 \text{ cm}^2 \text{ V}^{-1} \text{ s}^{-1}$ and the 338 highest conductivity with a maximum value of 957 S cm^{-1} ." I believe the authors refer to neutral polymer when it comes to charge carrier mobility and doped polymer when it comes to conductivity.

Last, I also wonder whether the observations can be generalized in other polymers. Or I ask a different question. Can it be generalized? Would authors be able to use molecular dynamics and higher-level computational tools to visualize assembly behaviors? It might be a good direction to collaborate with others in this aspect. By no means, I am asking for this manuscript. It is challenging. Many of us in the community have thought about this for a long time.

Reviewer #3:

Remarks to the Author:

The development of high-performance conjugated polymers is a crucial aspect of organic electronics. However, studying their aggregation structures poses significant challenges due to the complexity of their structures and the inhomogeneity of their molecular weight distribution. In this manuscript, the authors used advanced techniques such as liquid-phase electron microscopy, absorption spectroscopy, atomic force microscopy, and grazing incidence X-ray diffraction to demonstrate how molecular weight affects the aggregation structure of isoindigo-bithiophene copolymers and its impact on carrier transport performance. The direct observation of conjugated polymer disaggregation using liquid-phase electron microscopy is particularly noteworthy. The demonstration of high conductivity achieved by conjugated polymers with narrow molecular weight distribution also underscores the importance of tuning their aggregated structure. Overall, this is an excellent manuscript. If the following minor issues can be resolved, I can recommend the manuscript for publication in Nature Communications:

1. The authors should briefly explain the mechanism behind the disaggregation of aggregated structures of conjugated polymers under liquid-phase electron microscopy. Additionally, it would be helpful to know if this disaggregation is reversible.
2. The authors mention the use of molecular dynamics simulations. Can the process of aggregate formation in conjugated polymers be visualized through these simulations? If not, what challenges make it difficult to do so?
3. Using (100) diffraction about the three small molecular compounds in the GIWAXS experiment is not entirely appropriate, as it cannot be compared with polymers.
4. To aid in comparing the magnitude of improvement, it would be helpful to present mobility and maximum conductivity data in logarithmic coordinates.
5. Supplementary Figure S33 must compare all three small molecule compounds, specifically their maximum ratio of A_P1 to A_N0. Additionally, it is essential to point out that the decrease in this ratio after an increase in molecular weight is universal, and the figure should include a brief discussion on this topic.
6. Supplementary Figures S1, S2, and S3 should include the yield of each step to provide a better

understanding of the synthesis process. Additionally, Figure 1 in the main text should directly mark the yield rate to highlight the challenges of obtaining these materials.

Point-to-Point Response

For Reviewer 1:

Comment: *This manuscript by Zhou, Y.-Y et al. demonstrates an analysis on assembled polymeric structures that depend on chain lengths of monomers, and relates the structures to functionality. The authors have performed experiments in multiple scales using various techniques including cryo-EM, graphene liquid-phase TEM, UV-vis, X-ray scattering, and charge transport measurements. Although the study is interesting and may be beneficial for understanding structural effects on performance for polymeric devices, the conclusions made in this study have already been established previously in other publications and do not seem to bring enough new insights in its current form. However, having following concerns fully addressed, the work can be recommended for a publication in Nature Comm.*

Our Response:

We thank the reviewer for reading our paper carefully and giving constructive suggestions. We would like to make the following replies about the question on the innovation of our manuscript:

(1) Our work provided a deep understanding of the inheritance of assemblies from solution-state aggregation to solid-state microstructures. In previous research, the high-density tie chains were thought to be the key to achieve excellent charge transport performance (*Nat. Mater.* 2013, **12**, 1038–1044). We clearly demonstrated the origin of high-density tie chains in films composed of high-molecular-weight polymers and the crucial effect of molecular weight (chain length) on the electrical performance of conjugated polymers.

(2) Our work innovatively adopted several advanced techniques to visualize an entire evolution venation of conjugated polymers from single chain, to solution-state aggregation, to film formation kinetics, to solid-state morphology, and finally to device electrical performances, which has never been accomplished in previous works.

(3) Considerable research efforts have been devoted to the influence of homopolymers' molecular weight on their solid-state microstructures and electrical performance, and poly(3-hexylthiophene) (P3HT) is a typical system (*Macromolecules* 2013, **46**, 9349–9358). Nevertheless, compared with P3HT, donor-acceptor (D-A) conjugated polymers exhibit larger-delocalized and enhanced backbone rigidity. Therefore, the transition of chain behavior with molecular weight gained from P3HT cannot be directly migrated to D-A materials (*Macromolecules* 2021, **54**, 8207–8219). However, the mechanism about the effect of molecular weight on D-A conjugated polymers still remains unclear (*Macromolecules* 2021, **54**, 10203–10215; *Adv. Mater.* 2022, **34**, 2108255). Our manuscript filled the gap in this section using a representative isoindigo-based D-A polymers.

Revision in manuscript:

Main text, Page 1, line 27: Visualizing multi-level assembly structures of conjugated polymers provides a deep understanding of the inheritance of assemblies from solution-state aggregation to solid-state microstructures, accelerating the optimization of solution processing and device fabrication.

Main text, Page 2, line 16: Herein, several advanced techniques were innovatively adopted to visualize an entire evolution venation of the multi-level assembly structures of conjugated polymers and study their structure-performance relationship.

Main text, Page 12, line 13: Our work reports the direct visualization of structural evolution of isoindigo-based conjugated molecular aggregates in solution, which filled the gap in the effect of chain-length dependent aggregation on the charge transport properties of D-A conjugated molecular system.

Main text, Page 12, line 16: It should be pointed out that the charge transport properties of conjugated molecular system vary by several orders of magnitude with molecular weight, thus it is very important to investigate and regulate chain length of conjugated polymers.

Question 1: *One suggestion may be to provide further quantitative analysis and solid conclusions on the results for the solution-phase observations of polymeric chains using TEM, which could bring about information into the dynamics that may govern the assembly or disassembly process. Some questions I had were: (1) Is there a relationship between the location of the bubble interface within the graphene liquid cell and the location of the polymeric disassembly onset? (2) How significant is the gas-liquid interface in actual solution processes, and how well can the graphene liquid cell, which is a confined environment, model the actual disassembly process that occurs in batch solution? The bubbles formed are radiolysis products, and will also have a different gas composition. (3) Are there changes in the dynamics of the separated polymer strands once disassembly occurs that could affect the overall assembly process?*

Our Response:

(1) We would like to thank the reviewer for their helpful note, which has enabled us to further elaborate on three key aspects relating to the location of the bubble interface and the disassembly process. Firstly, we believe that the onset of the disassembly process can be attributed to the surface tension and/or shear force at the gas-liquid interface, which overcomes the van der Waals forces arising from stacking in our system (*Adv. Mater.* 2022, **34**, 2202353). Such perturbation due to the presence of gas-liquid interface is ubiquitous in solution-based processing method as solvent evaporates. To investigate the dynamics of the disassembly process as a function of temperature, we plan to combine Heating In-situ Holders and graphene liquid cell in the near future. Secondly, regarding graphene liquid cells larger than 100 nm, most bubbles appear to occupy the entire pocket width (as described in *ACS Nano* 2022, **16**, 18526–18537). In this scenario, polymer strands either gather along the edge of the bubble (which is not recorded in our experiment), or are located on the gas-liquid interface. As a result, the bubbles are not site-specific, and neither are the polymer strands. Thirdly, since there is only one bubble in the graphene liquid cell and the disassemble process occurs at the gas-liquid interface, we further analyzed the differences between polymer strands located at the concave interface and in the middle of the bubble (as shown in Supplementary Fig. 15a). We observed that there were insignificant differences in the changes of projected area between the polymer strands located at the concave interface, leading us to believe that the disassemble process is not related to the location of the polymer strands at a single bubble in the liquid cell (Supplementary Fig. 15b). Therefore, we conclude that the dynamics we observed are independent of the location of the polymer strands and are instead governed by the local bubble size and curvature, which are directly related to the local stress applied to the polymer strands. We hope these clarify any doubts or questions regarding the role of bubble location in our study.

Revision in manuscript and Supplementary Information:

Main text, Page 5, line 11: The surface tension and the shear force at the gas-liquid interface provide a sufficient driven force for the disaggregation of aggregates⁴² (Supplementary Note 3).

Main text, Page 5, line 27: The increase in the projected area and perimeter coincides with the splitting process of thick fiber, indicating the disaggregation process (Supplementary Fig. 9, Fig. 11, Fig. 14 and Fig. 15 and Supplementary Note 4).

Supplementary Information:

Supplementary Figure 15 (a) Image of a bubble in the liquid cell. The location of concave surface and middle of the bubble were marked by arrows. **(b)** Projected sizes versus time of the aggregate at middle of bubble and concave surface.

(2) Indeed, the effect of surface interaction, confinement, and radiolysis in the liquid cell should be considered when interpret for the observed dynamics. As the reviewer noted, the sluggish motion of nanoparticles and molecules in LP-EM compared to their behavior in bulk solution can be advantageous in capturing and analyzing intermediate states during a dynamic process. The presence of the confinement and surface interactions in the liquid cell effectively retard the kinetics of the system such that it acts as a “slow motion” camera, allowing to capture the intermediate states that are too transient to capture in bulk solution. However, it is important to carefully consider the potential artifacts of confinement and surface interactions on the interpretation of the observed behavior. The conclusions drawn from LP-EM experiments were carefully correlated with those from other experiments conducted in bulk solution. The gas-liquid interface plays a critical role in the film formation using solution processing method, which has a profound impact on the inheritance of solution-state aggregation into solid-state microstructures. This has been discussed in the main text (refer to Page 8, Line 9) and demonstrated by *in situ* UV-*vis* absorption spectra, AFM, and EM. Besides, the results analyzed from the temperature-dependent absorption spectra about the disaggregation process were also self-consistent with the results of LP-EM. With the conclusion from other analytical tools, we believe that the data generated by LP-EM experiments under current experimental conditions can reflect the disassembly process that occurs in batch solution as realistically as possible. Furthermore, it is noteworthy that the disassembly process induced by the liquid-gas interface, despite potential kinetic disparities in bulk solution, still serves as a representative model for comprehending the mechanisms of film formation. In this context, film formation involves interfacial phenomena, including surface tension and shear forces during the process of evaporation. Though the gas composition of the bubble, predominantly comprised of hydrogen and oxygen, varies from that of the air, which is typically composed of oxygen and nitrogen, polymers in this work are inert in both environments. Hence, we propose that the findings ascertained via LP-EM experiments conducted under the prevailing experimental conditions can reflect the disassembly process occurring in a batch solution.

(3) We thank the reviewer for this helpful suggestion. We did observe that larger polymer aggregates tend to disassemble into several smaller aggregates, which also has a possibility to reaggregate again. The dispersed chains may tend to aggregate or form different types of structures, which could ultimately affect the final structure and properties of the assembled polymers. This dynamic process is illustrated in the Supplementary Fig. 12. Over time, small aggregates tend to accumulate if the gas-liquid interface gradually shrinks and the local concentration increases. Such findings hold important implications for our understanding of the assemble and disassemble process of polymers and highlight the importance of

continued investigation into tuning these processes through temperature or concentration control. The ability to manipulate these processes could have significant impacts on the structural and functional properties of the assembled polymer film, including its mechanical and electrical properties, solubility as well as stability. We also plan to combine Heating In-situ Holders and graphene liquid cell to further investigate the ideal dynamics of the unidirectional disaggregation process of conjugated polymers.

Revision in manuscript and Supplementary Information:

Main text, Page 5, line 23: A similar process was also captured in repeated experiments (Movie S2, Supplementary Fig. 10, Fig. 11, and Fig. 12)

Supplementary Information:

Supplementary Figure 12 Extracted images from Movie S2, which exhibited a reaggregate process of polymer strands. Scale bar: 20 nm.

Question 2: The authors should additionally provide information on the beam dose rate and the threshold electron dose of the polymers, and comment on how tolerant the polymers are to the electron beam.

Our Response:

Undoubtedly, concerns remain toward the non-invasiveness of electron probe to carbon-based macromolecules. The expected critical dose in liquids is theoretically presumed to be approximately $1000 \text{ e}^- \cdot \text{\AA}^{-2}$, about two orders of magnitude higher than that in Cryo-EM (approximately $100 \text{ e}^- \cdot \text{\AA}^{-2}$ for protein structure and approximately $10 \text{ e}^- \cdot \text{\AA}^{-2}$ for active enzyme structure (*Nat. Rev. Mater.* 2019, **4**, 61–78). This can be attributed to the more diffusive and less reactive radicals present, as well as the radical scavenging ability of graphene (*Nano Lett.* 2017, **17**, 414–420). This estimation is reasonable and supported by previous works concerning more fragile bio-macromolecule, which imaged DNA motion in graphene liquid cell (GLC) (*Nano Lett.* 2013, **13**, 4556–4561; *Proc. Natl. Acad. Sci.* 2020, **117**, 1283–1292) and observed intactness of whole mammalian cells in a liquid (*Nat. Rev. Mater.* 2019, **4**, 61–78).

In our system, the beam dose rates of Movie S1, S2, and S3 were 14.5, 9.6, and $3.9 \text{ e}^- \cdot \text{\AA}^{-2} \cdot \text{s}^{-1}$. Experimentally, we observed that the damage to polymer strands manifests as severe scission and a sharp decrease in projection area within a span of approximately two seconds (blue area in Supplementary Fig. 16), which is consistent with common degradation patterns in GLCs (*ACS Nano* 2022, **16**, 18526–18537). At 157.6 s, we noted the onset of severe scission when the electron dose reached approximately $2283 \text{ e}^- \cdot \text{\AA}^{-2}$, followed by a marked reduction in projected area at 173.8 s. Based on the apparent disruption of assemble structure, we determined the critical dose to be around $2000 \text{ e}^- \cdot \text{\AA}^{-2}$. We also assessed the time and electron dose before the appearance of bubbles in five additional liquid cells that contained identical solvents. The total electron dose before bubbles appeared ranged between 966 and $4793 \text{ e}^- \cdot \text{\AA}^{-2}$, leading us to conclude that our polymers can withstand at least $1000 \text{ e}^- \cdot \text{\AA}^{-2}$. Furthermore, we assessed the total electron dose of Movies S1, S2, and S3, which were 116, 557, and $59 \text{ e}^- \cdot \text{\AA}^{-2}$, respectively, and concluded that all analyzed polymer strands remained intact. Our experiment was conducted with the objective of avoiding artifacts resulting from electron beam damage, and we hope

that the potential perturbation of the electron beam to its native dynamics will be better defined in near future.

Revision in manuscript and Supplementary Information:

Main text, Page 5, line 29: All analyzed samples remained intact under the used electron dose (Supplementary Fig. 16 and Supplementary Note 5).

Main text, **Methods: Liquid transmission electron microscope (LP-EM):** These electron doses were within the typical ranges used for organic materials, under which structures remain intact^{46,57,58}.

Main text, **Reference:**

57. de Jonge, N., Houben, L., Dunin-Borkowski, R. E. & Ross, F. M. Resolution and aberration correction in liquid cell transmission electron microscopy. *Nat. Rev. Mater.* **4**, 61–78 (2019).

58. Gibson, W. & Patterson, J. P. Liquid Phase Electron Microscopy Provides Opportunities in Polymer Synthesis and Manufacturing. *Macromolecules* **54**, 4986–4996 (2021).

Supplementary Information:

Supplementary Figure 16. (a) Representative images of the damage process of polymer strands under continuous electron beam irradiation. (b) Projected size versus time of this process. Our polymer can withstand at least 1000 e⁻ · Å⁻². We assessed the total electron dose of supporting movies S1, S2, and S3, which were 116, 557, and 59 e⁻ · Å⁻², respectively, and concluded that all analyzed polymer strands remained intact.

Question 3: Maybe the authors could use other liquid cell designs to capture the dissolution process of the longer polymers, as there are a variety of liquid cells available. Using a spacer within the graphene sandwiches may allow a bigger sized liquid pocket.

Our Response:

To overcome the limitations of graphene liquid cells for longer polymer strands, we opted for the use of a commercially available silicon nitride liquid cell. Unfortunately, its spatial resolution is inadequate to enable the identification of singular polymer strands. Hopefully, employing a spacer within the graphene sandwiches can offer a larger space for longer polymers (*Nano Lett.* 2018, **18**, 1168–1174; *Adv. Mater.* 2020, **32**, 2002889; *Nano Lett.* 2022, **22**, 7423–7431). However, the sample preparation technology of this kind of liquid cell has not been commercialized by now. We are optimistic that this innovative liquid cell design will soon be applicable to polymer systems, as currently it has primarily been tested with inorganic nanoparticles.

Question 4: What is the gray shaded area in figures 2(e-i)? We noticed that there is also a gray 'shadow' next to the legend in figure 2. Also, perhaps reorganize figures 2(e-i) so that they are separated from

figure 2(d), as the color scheme of the two sets of figures are similar (while one is for polymer type and one is for different temperatures) and may cause confusion.

Our Response:

The gray shaded area was marked to highlight the onset variation regions of these spectra, which were ranging from 700 – 760 nm. To avoid confusion, we changed the gray shaded rectangles to small red dotted rectangles and added corresponding explanation in the figure legend. The gray ‘shadow’ next to the legend was a part of frame of the legend. We deleted it in our revised manuscript to eliminate misunderstanding. Accordingly, we changed the color scheme of Figures 2(e-i) to distinguish the details of Figure 2d and Figures 2(e-i).

Revision in manuscript:

Fig. 2. Visualizing the dynamic disaggregation process of polymers in solutions. a. Schematic of sample preparation of graphene liquid pockets for *in situ* LP-EM study. b. LP-EM images for the dynamic disaggregate process of IIDDT₁₉ in a liquid pocket, at the bubble/liquid interface, mimicking the condition for solution processing of conjugated polymer. The red marks identifying the image-tracking results. Scale bar: 20 nm. The solution concentration is 1 g·L⁻¹. c. The corresponding binarized images obtained from threshold intensity. d. Temperature-dependent absorption spectra showing the optical bandgap variation as a function of temperature for the sample oligomers and polymers. e-i. Temperature-dependent absorption spectra of 0.01 g·L⁻¹ oligomers and polymers in 1-chloronaphthalene (CN). The red dotted rectangles marked the onset variation regions of these spectra.

Question 5: Additionally, there were several grammatical errors that must be corrected in the manuscript. Some I highlight are:

- a. "In the contrast..." (Line 67) should be "In contrast" or "On the contrary".
- b. "longer polymer chains owing to its flexible conformation..." (Line 68) should be "to their" instead of "its"
- c. "long polymer chains with a broad dispersity were usually used..." (line 79) should be "were used" if you used it in this experiment.

Our Response:

We have thoroughly checked our manuscript and corrected the grammatical errors and typos, which were highlighted in our revised manuscript.

Revision in manuscript:

Main text, Page 2, line 26: On the contrary, long polymer chains owing to their flexible conformation generate interconnected aggregates network in solution and directly form less crystalline interconnected microstructures with high mobility during drying.

Main text, Page 2, line 38: Long polymer chains with a broad dispersity were used as a blended system exhibiting homogeneous properties^{44,45}.

For Reviewer 2:

Comment: Charge transport behaviors of conjugated polymers are complex, arising from the interplay of structural and morphological influence of polymer chains. Due to the polydispersity nature of polymer chains, it is rather difficult to disentangle this influence. This manuscript is beautifully carried out and attempts to reveal charge transport behaviors by revealing the assembly structures. This is made possible by carefully fractioning polymer chains. What is appealing in this study is the virtualization of chain-length dependent assembly behaviors. Many of the observations have been known and demonstrated, such as crystalline structures from short chains and tie-chain connections from long chains. In a previous study, Chaudhry et al even made well-defined discrete oligomers to illustrate chain dynamics. Salleo, Diao, and many others have done nice work to help understand assembly behaviors. This study is kinda like the Jewel in the Crown and elegantly pieces important aspects together to provide a whole picture. In my opinion, it will be well received in the community.

Our Response:

We thank the reviewer for the strong support of our manuscript.

Question 1: There is confusion in the conclusion "Hence, IIDDT₉₉, the polymer with long and flexible chain conformation, exhibited the highest carrier mobility up to 3.23 cm² V⁻¹ s⁻¹ and the 338 highest conductivity with a maximum value of 957 S cm⁻¹." I believe the authors refer to neutral polymer when it comes to charge carrier mobility and doped polymer when it comes to conductivity.

Our Response:

We have corrected this sentence and thoroughly checked and revised our manuscript to avoid confusing expressions.

Revision in manuscript:

Main text, Page 12, line 9: Hence, IIDDT₉₉, the polymer with long and flexible chain conformation, exhibited the highest carrier mobility up to 3.23 cm² V⁻¹ s⁻¹ on neutral state and the highest conductivity with a maximum value of 957 S cm⁻¹ after doped.

Question 2: Last, I also wonder whether the observations can be generalized in other polymers. Or I ask a different question. Can it be generalized? Would authors be able to use molecular dynamics and

higher-level computational tools to visualize assembly behaviors? It might be a good direction to collaborate with others in this aspect. By no means, I am asking for this manuscript. It is challenging. Many of us in the community have thought about this for a long time.

Our Response:

It is really significant as reviewer suggested to develop the generalization of our observations in other polymers. Recently, Wang et al. also investigated a series of diketopyrrolopyrrole (DPP) based polymers (TDPP-Se) with different molecular weights. They found polymers with low molecular weights showed fibrillar aggregates with short lengths, while high-molecule-weight polymers exhibited long fibrillar aggregates (*Adv. Mater.* 2022, **34**, 2108255). Kline and co-workers found P3HT films with low molecular weight had a highly ordered structure composed of nanorods. For films with high molecular weight, long chains bridge the ordered regions and soften the boundaries (*Macromolecules* 2005, **38**, 3312-3319). Noriega et al. also proved that short-range intermolecular aggregation is sufficient long-range charge transport in many conjugated polymers, including poly(2,5-bis(3-tetradecylthiophen-2-yl)thieno[3,2-b]thiophene) (PBTTT), poly[5,5'-bis(3-alkyl-2-thienyl)-2,2'-bithiophene] (PQT), indacenodithiophene-based polymers (IDT-BT), and so on (*Nat. Mater.* 2013, **12**, 1038–1044). These literatures proved that our observations in the IID-based oligomers and polymers, which mainly consisted of the differences of solution-state aggregation and solid-state morphology, were similar with many other conjugated polymer systems.

Moreover, molecular dynamics simulation is an important method to understand the physical and chemical process of complex systems. However, it's difficult to visualize assembly behaviors of these polymers in solutions through molecular dynamics simulation. First, chain aggregation occurs only at very long-time scales, well beyond what is currently computationally feasible in terms of molecular dynamics simulation (*J. Mater. Chem. C* 2018, **6**, 13162-13170). Second, the computational objects of molecular dynamics were usually oligomers with length no longer than 10 repeat-units (*Angew. Chem. Int. Ed.* 2021, **60**, 20483; *Angew. Chem. Int. Ed.* 2020, **59**, 17467; *J. Mater. Chem. C* 2019, **7**, 14198). The lengths of these polymers in our system were too large to be simulated.

Although it is difficult to simulate the assembly behaviors of polymers, we tried to simulate the disaggregation process of an oligomer with six repeating units (as shown in Supplementary Fig. 18 and Fig. 19). Within the operation range of molecular dynamics simulation, we simulated the disaggregation process of (IID-DT)₅-IID (which has been proven to start showing similar characteristics with polymers). At 25 °C, the initial aggregate got swollen and showed a tendency to separate into two smaller aggregates, but the whole aggregate maintained long-range order. At 100 °C, the aggregate split into two 4-chain and 2-chain aggregates. The tendency of above simulated results was consistent with the analysis of the temperature-dependent absorption spectra and LP-EM.

Revision in manuscript and Supplementary Information:

Main text, Page 6, line 12: Besides, molecular dynamics simulations were also performed on (IID-DT)₅-IID to qualitatively analyze the disaggregated process. At 25 °C, the initial aggregate got swollen and showed a tendency to separate into two smaller aggregates, but the whole aggregate maintained long-range order. At 100 °C, the aggregate split into two 4-chain and 2-chain aggregates (Supplementary Fig. 18 and Fig. 19). The tendency of the simulated results was consistent with our experimental observations^{33,48}.

Main text, **Methods: Molecular dynamics simulations.** All-atom molecular dynamics simulations were performed using Gromacs-2019.4 with GPU acceleration⁵⁹. General AMBER force field (GAFF) was adopted to describe the interactions between atoms⁶⁰. Atomic charges were calculated from the restrained

electrostatic potentials (RESP) at the ω B97XD/6-31G(d) level of theory using Multiwfn program^{61,62}. The initial models for the molecular dynamics simulations were built by constructing oligomer stacks (consisting of six oligomer chains perfectly packed on top of each other) (Supplementary Fig. 18); each oligomer chain is 6 repeat-units long. The simulation boxes consisted of 102708 atoms in total, including 8000 oDCB molecules. The molecular dynamics simulations were performed at 25 °C (for 20 ns) or 100 °C (for 20 ns).

Main text, **Reference:**

33. Ning, L., Han, G. & Yi, Y. Conformational and aggregation properties of PffBT4T polymers: Atomistic insight into the impact of alkyl-chain branching positions. *J. Mater. Chem. C* **7**, 14198–14204 (2019).

48. Ashokan, A., Wang, T., Ravva, M. K. & Brédas, J. L. Impact of solution temperature-dependent aggregation on the solid-state packing and electronic properties of polymers for organic photovoltaics. *J. Mater. Chem. C* **6**, 13162–13170 (2018).

59. Abraham, M. J. *et al.* GROMACS: High performance molecular simulations through multi-level parallelism from laptops to supercomputers. *SoftwareX* **1–2**, 19–25 (2015).

60. Wang, J., Wolf, R. M., Caldwell, J. W., Kollman, P. A. & Case, D. A. Development and testing of a general Amber force field. *J. Comput. Chem.* **25**, 1157–1174 (2004).

61. Tian, L. & Fei-Wu, C. Comparison of Computational Methods for Atomic Charges. *Acta Physico-Chimica Sin.* **28**, 1–18 (2012).

62. Lu, T. & Chen, F. Multiwfn: A multifunctional wavefunction analyzer. *J. Comput. Chem.* **33**, 580–592 (2012).

Supplementary Information:

Supplementary Figure 18. Illustration of the initial models used for the molecular dynamics simulations. Top: A single polymer chain of (IID-DT)₅-IID. Bottom: A polymer aggregate made of 6 polymer chains perfectly packed on top of each other.

Supplementary Figure 19. (a) Initial model parallel π - π stacked polymer chains (top) and equilibrium snapshot taken from the molecular dynamics simulations at 25 °C. (b) Initial model parallel π - π stacked polymer chains (top) and equilibrium snapshot taken from the molecular dynamics simulations at 100 °C.

For Reviewer 3:

Comment: The development of high-performance conjugated polymers is a crucial aspect of organic electronics. However, studying their aggregation structures poses significant challenges due to the complexity of their structures and the inhomogeneity of their molecular weight distribution. In this manuscript, the authors used advanced techniques such as liquid-phase electron microscopy, absorption spectroscopy, atomic force microscopy, and grazing incidence X-ray diffraction to demonstrate how molecular weight affects the aggregation structure of isoindigo-bithiophene copolymers and its impact on carrier transport performance. The direct observation of conjugated polymer disaggregation using liquid-phase electron microscopy is particularly noteworthy. The demonstration of high conductivity achieved by conjugated polymers with narrow molecular weight distribution also underscores the importance of tuning their aggregated structure. Overall, this is an excellent manuscript. If the following minor issues can be resolved, I can recommend the manuscript for publication in *Nature Communications*.

Our Response:

We thank you so much for your positive and constructive comments of our manuscript.

Question 1: The authors should briefly explain the mechanism behind the disaggregation of aggregated structures of conjugated polymers under liquid-phase electron microscopy. Additionally, it would be helpful to know if this disaggregation is reversible.

Our Response:

According to literature, we believe that the onset of the disassembly process can be attributed to the surface tension and/or shear force at the gas-liquid interface, which overcomes the van der Waals forces arising from stacking in our system (*Adv. Mater.* 2022, **34**, 2202353). Of all the videos we have recorded, all disassemble processes (Movie S1, S2, S3) appear to occur at the interface. However, if the gas-liquid

interface gradually shrinks and the local concentration increases, we think that the chains have a possibility to reaggregate, as shown in Supplementary Fig. 12. Ideally, if we want to investigate the unidirectional disaggregation process of conjugated polymers in pure solution (not gas-liquid interface), Heating In-situ Holders and graphene liquid cell with engorged liquid pocket should be combined to take the experiment, and efforts are currently underway in our laboratory.

Revision in manuscript and Supplementary Information:

Main text, Page 5, line 11: The surface tension and shear force at the gas-liquid interface provide a sufficient driven force of the disaggregation of aggregates⁴² (Supplementary Note 3).

Supplementary Information:

Supplementary Figure 12 Extracted images from Movie S2, which exhibited a reaggregate process of polymer strands. Scale bar: 20 nm.

Question 2: The authors mention the use of molecular dynamics simulations. Can the process of aggregate formation in conjugated polymers be visualized through these simulations? If not, what challenges make it difficult to do so?

Our Response:

Molecular dynamics simulation is an important method to understand lots of physical and chemical process in complex systems. However, in our system, it's difficult to visualize the process of aggregate formation in conjugated polymers through this method. First, chain aggregation occurs only at very long-time scales, well beyond what is currently computationally feasible in terms of molecular dynamics simulation (*J. Mater. Chem. C* 2018, **6**, 13162-13170). Second, the computational objects of molecular dynamics were usually oligomers with length no longer than 10 repeat-units (*Angew. Chem. Int. Ed.* 2021, **60**, 20483; *Angew. Chem. Int. Ed.* 2020, **59**, 17467; *J. Mater. Chem. C* 2019, **7**, 14198). The lengths of these polymers in our system were too large to be simulated to visualize the dynamics aggregation or disaggregation process.

However, we simulated the disaggregation process of (IID-DT)₅-IID (which has been proven to start showing similar characteristics with polymers) within the operation range of molecular dynamics simulation (as shown in Supplementary Fig. 18 and Fig. 19). As consistent with the tendency of temperature-dependent absorption spectra and LP-EM, at 25 °C, the initial aggregate got swollen and showed a tendency to separate into two smaller aggregates, but the whole aggregate maintained long-range order; at 100 °C, the aggregate split into two 4-chain and 2-chain aggregates.

Revision in manuscript and Supplementary Information:

Main text, Page 6, line 12: Besides, molecular dynamics simulations were also performed on (IID-DT)₅-IID to qualitatively analyze the disaggregated process. At 25 °C, the initial aggregate got swollen and showed a tendency to separate into two smaller aggregates, but the whole aggregate maintained long-range order. At 100 °C, the aggregate split into two 4-chain and 2-chain aggregates (Supplementary Fig. 18 and Fig. 19). The tendency of the simulated results was consistent with our experimental observations^{33,48}.

Main text, **Methods: Molecular dynamics simulations.** All-atom molecular dynamics simulations were performed using Gromacs-2019.4 with GPU acceleration. General AMBER force field (GAFF) was adopted to describe the interactions between atoms. Atomic charges were calculated from the restrained electrostatic potentials (RESP) at the ω B97XD/6-31G(d) level of theory using Multiwfn program. The initial models for the molecular dynamics simulations were built by constructing oligomer stacks (consisting of six oligomer chains perfectly packed on top of each other) (Supplementary Fig. 18); each oligomer chain is 6 repeat-units long. The simulation boxes consisted of 102708 atoms in total, including 8000 *o*DCB molecules. The molecular dynamics simulations were performed at 25 °C (for 20 ns) or 100 °C (for 20 ns).

Main text, **Reference:**

33. Ning, L., Han, G. & Yi, Y. Conformational and aggregation properties of PffBT4T polymers: Atomistic insight into the impact of alkyl-chain branching positions. *J. Mater. Chem. C* **7**, 14198–14204 (2019).

48. Ashokan, A., Wang, T., Ravva, M. K. & Brédas, J. L. Impact of solution temperature-dependent aggregation on the solid-state packing and electronic properties of polymers for organic photovoltaics. *J. Mater. Chem. C* **6**, 13162–13170 (2018).

Supplementary Information:

Supplementary Figure 18. Illustration of the initial models used for the molecular dynamics simulations. Top: A single polymer chain of (IID-DT)_s-IID. Bottom: A polymer aggregate made of 6 polymer chains perfectly packed on top of each other.

Supplementary Figure 19. (a) Initial model parallel π - π stacked polymer chains (top) and equilibrium snapshot taken from the molecular dynamics simulations at 25 °C. (b) Initial model parallel π - π stacked polymer chains (top) and equilibrium snapshot taken from the molecular dynamics simulations at 100 °C.

Question 3: Using (100) diffraction about the three small molecular compounds in the GIWAXS experiment is not entirely appropriate, as it cannot be compared with polymers.

Our Response:

We have corrected the corresponding Supplementary Figures in our revised manuscript.

Revision in manuscript and Supplementary Information:

Main text, Page 10, line 15: As the chain length further increased, the (h00) diffraction peaks showed decreased coherence length and increscent paracrystallinity disorder (Supplementary Fig. 29). These results indicated that polymers with longer chains formed films of lower crystallinity.

Supplementary Information:

Supplementary Figure 27. GIWAXS patterns of pristine (a) (IID-DT)₂-IID, (b) (IID-DT)₃-IID, and (c) (IID-DT)₅-IID films.

Supplementary Figure 29. GIWAXS analysis on coherence length of (100) diffractions of pristine films. GIWAXS analysis on paracrystalline disorder of (100) diffractions of pristine films.

Supplementary Figure 33. GIWAXS patterns of (a) (IID-DT)₂-IID, (b) (IID-DT)₃-IID, (c) (IID-DT)₅-IID, (d) IIDDT₁₉, (e) IIDDT₅₇, and (f) IIDDT₉₉ films doped by FeCl₃.

Supplementary Figure 35. GIWAXS analysis on coherence length of (100) diffractions of doped films. GIWAXS analysis on paracrystalline disorder of (100) diffractions of doped films.

Question 4: To aid in comparing the magnitude of improvement, it would be helpful to present mobility and maximum conductivity data in logarithmic coordinates.

Our Response:

We changed the mobility and maximum conductivity data in logarithmic coordinates.

Revision in manuscript:

Fig. 5. Electrical performance of oligomers and polymers. a. The dependence of mobility on the chain length. Inset: Diagram of a polymer transistor. Due to the limitation of solution viscosity, the FET devices of (IID-DT)₂-IID and (IID-DT)₃-IID were fabricated with 20 mg mL⁻¹ trichloro ethylene solutions. b. Maximal electrical conductivity of the samples doped by FeCl₃. c. Electrical conductivities of these polymers doped for 5, 10, 30, 60, 300, and 600 s, respectively. d-f. UV-vis-NIR absorption spectra of these polymer films plotted as a function of time (0, 5, 10, 30, 60, 300, and 600 s) with doping (The break between 860 nm-880 nm was ascribed to the fluctuation of the instrument.).

Question 5: Supplementary Figure S33 must compare all three small molecule compounds, specifically their maximum ratio of A_{P1} to A_{N0}. Additionally, it is essential to point out that the decrease in this ratio after an increase in molecular weight is universal, and the figure should include a brief discussion on this topic.

Our Response:

We supplied the data of oligomers in Supplementary Figure S38 as reviewer suggested. As it could be seen, the ratios of A_{P1}/A_{N0} decreased with polymers' molecular weight increased. The variation might be attributed the differences of solid-state structures of these materials. High crystallinity is usually accompanied with rough morphology in conjugated molecular system. The coarse morphology of oligomers created lots of cracks for dopants to permeate into the whole film and react with molecules (Supplementary Fig. 26 and 32). However, with molecular weight increasing, the film morphology became more uniform and more compact with fewer grain boundaries, which prevented dopants from efficiently penetrating into the film so that decreased the doping efficacy (Supplementary Fig. 26 and 32). Besides, we also added a brief discussion on the above topic to the legend of Supplementary Fig. 38.

Revision in manuscript and Supplementary Information:

Main text, Page 11, line 33: (A_{P1}: the absorbance of P1 peak; A_{N0}: the intensity of neutral peak in pristine film) (Supplementary Fig. 38). For (IID-DT)₂-IID and (IID-DT)₃-IID, the high doping efficiency might

be ascribed to the coarse morphology of film, which created lots of cracks for dopants to permeate and react with molecules.

Supplementary Information:

Supplementary Figure 26. AFM height images of (a) (IID-DT)₂-IID, (b) (IID-DT)₃-IID, (c) (IID-DT)₅-IID, (d) IIDDT₁₉, (e) IIDDT₃₇, and (f) IIDDT₉₉ films spin-coated by 5g L⁻¹ *o*DCB solutions.

Supplementary Figure 32. AFM height images of (a) (IID-DT)₂-IID, (b) (IID-DT)₃-IID, (c) (IID-DT)₅-IID, (d) IIDDT₁₉, (e) IIDDT₃₇, and (f) IIDDT₉₉ films doped by FeCl₃.

Supplementary Figure 38. The ratios of A_{P1}/A_{N0} evaluated from the *Vis*-NIR absorption spectra of these doped oligomers and polymers. For (IID-DT)₂-IID, the N_0 peak was at 680 nm, the P_1 peak was at around 790 nm; For (IID-DT)₃-IID, the N_0 peak was at 699 nm, the P_1 peak was at around 833 nm; For (IID-DT)₅-IID and these three polymers, the N_0 peak was at 713 nm, the P_1 peak was at around 900 nm; A_{P1} was the absorbance of P_1 peak and A_{N0} was the intensity of neutral peak at 713 nm in pristine film. The coarse morphology of oligomers created lots of cracks for dopants to permeate into the whole film and react with molecules. However, with molecular weight increasing, the film morphology became more uniform and more compact with fewer grain boundaries, which prevented dopants from efficiently penetrating into the film so that decreased the doping efficacy.

Question 6: *Supplementary Figures S1, S2, and S3 should include the yield of each step to provide a better understanding of the synthesis process. Additionally, Figure 1 in the main text should directly mark the yield rate to highlight the challenges of obtaining these materials.*

Our Response:

We marked the yield of each step in Supplementary Fig. 1, Fig. 2, and Fig. 3. We also gave the total yield of three oligomers and purification yield of these three polymers in Figure 1.

Revision in manuscript and Supplementary Information:

Fig. 1. Different chain conformations of IID-based oligomers and polymers. **a.** Chemical structures of three IID-based oligomers and three polymers of different molecular weights. **b.** Schematic of the synthesis process of three oligomers. **c.** Schematics of the purification process of these three polymers and the different evolution of the corresponding multi-level assembly structures. CF: chloroform; CB: chlorobenzene; *o*DCB: *o*-dichlorobenzene. **d.** The average distribution of fiber length counted from cryo-EM images of $(\text{IID-DT})_2\text{-IID}$, $(\text{IID-DT})_3\text{-IID}$, $(\text{IID-DT})_5\text{-IID}$, IIDDT₁₉, IIDDT₅₇, and IIDDT₉₉. The red line shows the calculated length of a single chain with the number of monomers. Insets: representative cryo-EM images.

Supplementary Information:

Supplementary Figure 1. Synthesis of (IID-DT)₂-IID.

Supplementary Figure 2. Synthesis of (IID-DT)₃-IID.

Supplementary Figure 3. Synthesis of (IID-DT)₅-IID.

Reviewers' Comments:

Reviewer #1:

Remarks to the Author:

The authors have addressed my questions regarding polymer dynamics in the graphene liquid cell, in terms of the effects of electron beam dose and the effect of the bubble interface. They have also addressed some of the limitations. This work is recommended for publication in Nature Communications.

Reviewer #2:

Remarks to the Author:

The authors have addressed my questions sufficiently.

Reviewer #3:

Remarks to the Author:

The revised manuscript has fully addressed the reviewers' comments. Overall, it is an excellent piece of work, and I highly recommend its publication without any further changes.